# Sensitivity of surface solar radiation to aerosol-radiation and aerosol-cloud interactions over Europe in WRFv3.6.1 climatic runs with fully interactive aerosols

Sonia Jerez[1*], Laura Palacios-Peña[1], Claudia Gutiérrez[2], Pedro Jiménez-Guerrero[1], Jose María López-Romero[1], E. Pravia-Sarabia[1], Juan Pedro Montávez[1]

(1) Department of Physics, Regional Atmospheric Modeling group, Regional Campus of International Excellence "Campus Mare Nostrum", University of Murcia, 30100 Murcia, Spain

(2) Environmental Sciences Institute, University of Castilla-La Mancha, 45071 Toledo, Spain

*Correspondence to: sonia.jerez@gmail.com

## Abstract

The amount of solar radiation reaching the Earth's surface can be highly determined by atmospheric aerosols, pointed as the most uncertain climate forcing agents through their direct (scattering and absorption), semi-direct (absorption implying a thermodynamic effect on clouds) and indirect (cloud properties modification when aerosols act as cloud condensation nuclei) effects. Nonetheless, Regional Climate Models hardly ever dynamically model the atmospheric concentration of aerosols and their interactions with radiation (ARI) and clouds (ACI). The objective of this work is to evince the role of modeling ARI and ACI in Weather Research and Forecast (WRF) model simulations with fully interactive aerosols (online resolved concentrations) with a focus on summer mean surface downward solar radiation (RSDS) over Europe. Under historical conditions (1991-2010), both ARI and ACI reduce RSDS by a few percentage points over central and northern regions. This reduction is larger when only ARI are resolved, while ACI counteract the effect of the former by up to half. The response of RSDS to the activation of ARI and ACI is mainly led by the aerosol effect on the cloud coverage, while the aerosol effect on the atmospheric optical depth plays a very minor role, which evinces the importance of the semi-direct and indirect aerosol effects. In fact, differences in RSDS among experiments with and without aerosols are softer under clear-sky conditions. In terms of future projections (2031-2050 vs. 1991-2010), the baseline pattern (from an experiment without aerosols) shows positive signals southward

and negative signals northward. While ARI enhance the former and reduce the latter, ACI work in
the opposite direction and provide a flatter RSDS change pattern, further evincing the opposite
impact from semi-direct and indirect effects and the non-banal influence of the latter.

## 1 – Introduction

Regional Climate Models (RCMs) are powerful tools providing high-resolution climate information
by dynamically downscaling coarser datasets, e.g. from Global Circulation Models (GCMs). Their
added value comes not only from the increased resolution, but also from the fact that such an
increased resolution allows modeling and considering fine scale processes and features that are
missed or misrepresented otherwise, e.g. local circulations and land uses (Rummukainen 2010,
Jacob et al 2014, 2020, Schewe et al 2019). Still, certain phenomena need to be parametrized, e.g.
the turbulence within the planetary boundary layer, microphysics processes and convective
phenomena. However, there are relevant processes that GCMs usually model dynamically, but
which are not usually included in RCMs runs. This is the case of the atmospheric aerosol
concentration and their multiple non-linear interactions (e.g. Taylor et al 2012 *vs.* Ruti et al 2016),
the so-called aerosol-radiation and aerosol-cloud interactions (ARI and ACI respectively; Boucher

43 2015).

Depending on their nature and the ambient conditions, aerosols can act to scatter and/or absorb the
solar radiation through ARI, which may result in less or more solar radiation reaching the surface
through direct and semi-direct effects. Direct effects might involve that less solar radiation reaches
the surface due to its scattering and absorption (Giorgi et al 2002, Nabat et al 2015a, Li et al 2017,
Kinne 2019), or more if, for instance, absorption warms aloft atmospheric layers, thereby leading to
more stable atmospheric situations (lower surface temperatures than upward) and thus to the
inhibition of clouds formation via convective phenomena (Giorgi et al 2002, Nabat et al 2015a).
Absorption itself can also lead to clouds inhibition and/or burn-off through thermodynamic effects,
i.e. by heating the air (semi-direct effects), thus increasing the amount of solar radiation reaching
the surface (Allen and Sherwood 2010). Besides, aerosols act as cloud condensation nuclei (indirect
effect or ACI), which may also result in less or more solar radiation reaching the surface.
Abundance of cloud condensation nuclei rebounds on enhanced scattering via whitened clouds of
smaller drops with increased size and lifetime, and on drizzle suppression which reduces bellow-
cloud wet deposition processes (Seinfeld et al 2016, Kinne 2019). On the contrary, in-cloud aerosol
scavenging processes lead to out-of-cloud cleaner atmospheres (Croft et al 2012). All these

processes can potentially alter local and regional circulations, therefore impacting beyond the radiative balance (Kloster et al 2010, Wilcox et al 2013, Nabat et al 2014, Wang et al 2016, Pavlidis et al 2020).

In the current context of climate crisis, the scientific challenge is becoming twofold: (1) to gain a good understanding of the processes that occur in the atmosphere and of what will occur in the future, because this is crucial (IPCC 2013) in order (2) to advance effective measures both at global and regional scales (IPCC 2014). In particular, climate change mitigation strategies require low-carbon energies to grow rapidly in the coming decades (Rohrig et al 2019, IRENA 2019). This rapid transition of the energy sector towards renewable-powered decarbonized systems makes energy production, transmission and distribution increasingly sensitive to weather and climate variability (Bloomfield et al 2016, Collins et al 2018, Kozarcanin et al 2018, Jerez et al 2019). Thus, several works have been devoted to assessing this issue through the use of climate modeling tools. In particular, for the solar resource, Crook et al (2011), Gaetani et al (2014), Wild et al (2015) and Müller et al (2019) showed a generalized increase in Europe by making use of GCM simulations, while Jerez et al (2015), Gil et al (2019) and Tobin et al (2018) reported a different behavior, with RCM simulations projecting a slight general decrease in the amount of solar radiation reaching the surface over Europe.

From the previous literature, we point out here three key features that motivated the present work. First, the increasing use of RCM to evaluate the renewable energy resources and their supply potential (e.g. Jerez et al 2013, 2015, 2019, Gil et al 2019, Soares et al 2019, van der Wiel et al 2019). Second, the key role of aerosols regarding the accuracy of the simulated solar resource by climate models (Gaetani et al 2014, Nabat et al 2015b, Pavlidis et al 2020), particularly attributed to their direct and semi-direct effects, which would help to explain the aforementioned discrepancy between the GCM and RCM future projections (Boé et al 2020, Gutiérrez et al 2020). Third, none of the previous studies has so far dealt with the non-evident RCM sensitivity to interactively modeled atmospheric aerosol concentrations and the resulting aerosol-radiation and aerosol-cloud interactions in order to simulate the solar resource under historical and future climate scenarios.

Hence, our objective here is to shed light on the third point above by assessing the sensitivity of long-term RCM simulations to the inclusion of ARI and ACI using fully interactive (online diagnosed) aerosols. For this, we made use of a widely applied RCM, the Weather Research and Forecasting (WRF) model (Skamarock et al 2008) and its coupled form with Chemistry (WRF-

Chem; Grell et al 2005), to perform sets of historical (period 1991-2010) and future (period 2031-2050) simulations over Europe in three ways: (1) without including atmospheric aerosols, (2) with dynamic aerosols and aerosol-radiation interactions activated, and (3) with dynamic aerosols and both aerosol-radiation and aerosol-cloud interactions activated.

Section 2 describes experiments and methods; section 3 presents the results; the discussion and conclusions are provided in Section 4.

## 2 – Experiments, data and methods

### 2.1 – General description of the WRF simulations

We performed three experiments using the WRF model version 3.6.1 (Skamarock et al 2008; available at https://www.mmm.ucar.edu/weather-research-and-forecasting-model). In all cases, the simulated periods were 1991-2010 (historical) and 2031-2050 (future). Initial and boundary conditions were taken from GCM simulations: the r1i1p1 MPI-ESM-LR historical and RCP8.5-forced runs (Giorgetta et al 2012a,b; available at https://cera-www.dkrz.de) from the Coupled Model Intercomparison Project Phase 5 (CMIP5; https://pcmdi.llnl.gov/mips/cmip5/; Taylor et al 2012). The Representative Concentration Pathway RCP8.5 (Moss et al 2010) depicts the highest radiative forcing along the XXI century among all RCPs, with doubled $CO_2$, $CH_4$, and $N_2O$ concentrations by 2050 compared to the last record of the historical period. Both the observed (past) and estimated (future) temporal evolution of the concentration of these species was appropriately considered in the WRF executions (Jerez et al 2018).

The three experiments consisted of, and are named as:

BASE: aerosols are not considered in the simulations. No aerosol climatology is used, and no aerosol interactions are taken into account by the model. WRF-alone considers a constant number of cloud condensation nuclei (250 per $cm^3$, set in the model by default) to enable the formation of clouds.

ARI: aerosols are estimated online and aerosol-radiation interactions are activated in the model (both direct and semi-direct effects are included in the simulations).

ARCI: aerosols are estimated online and both aerosol-radiation and aerosol-cloud interactions are
activated in the model (direct, semi-direct and indirect effects are included in the simulations).
The WRF spatial configuration consisted of two one-way nested domains (Supp Fig 1). The inner
one (target domain) is an Euro-Cordex (https://www.euro-cordex.net/; Jacob et al 2014, 2020)
compliant domain covering Europe with a horizontal resolution of 0.44º in latitude and longitude.
The outer one has a horizontal resolution of 1.32º and covers the most important areas of Saharan
dust emission  as in Palacios-Peña et al 2019a. This configuration was necessary to generate and
include the information of the Saharan dust intrusions through the boundaries of our target domain
for the ARI and ARCI experiments, because the boundary conditions from the GCM do not provide
this information. In the vertical dimension, 29 unevenly spaced eta levels were specified in the two
domains, with more levels near the surface than upward, and the model top was set to 50 hPa. The
physics configuration of the WRF model consisted of the Lin microphysics scheme (Lin et al 1983),
the RRTM long- and short-wave radiative scheme (Iacono et al 2008), the Grell 3D ensemble
cumulus scheme (Grell 1993, Grell and Dévényi 2002), the University of Yonsei boundary layer
scheme (Hong et al 2006) and the Noah land surface model (Chen & Dudhia 2001, Tewari et al
2004). Boundary conditions from the GCM were updated every 6 hours, including the low
boundary condition for the sea surface temperature. Nudging was applied to the outer domain, but
not to the target domain.
**2.2 – Including aerosols in WRF**
To perform the ARI and ARCI experiments, we used the WRF model coupled with Chemistry
(WRF-Chem) version 3.6.1 (Grell et al 2005,  Chin et al 2002). WRF-Chem runs with GOCART
aerosol module (Ginoux et al 2001). This scheme includes five species, namely sulfate, mineral
dust, sea salt aerosol, organic matter and black carbon, and was coupled with RACM-KPP
(Stockwell et al 1997, Geiger et al 2003) as chemistry option. Chemical reactions in the GOCART
model include several oxidation processes by the three main oxidants in the troposphere: OH, $NO_3$,
and $O_3$. The OH radical dominates oxidation during the daytime, but at night its concentration drops
and $NO_3$ becomes the primary oxidant (Archer-Nicholls et al 2014). So, the oxidation pathways
represented in GOCART include: (a) dimethyl sulfide (DMS) oxidation by the hydroxyl radical
(OH) during the day to form sulfur dioxide ($SO_2$) and methanesulfonic acid (MSA); (b) oxidation
by nitrate radicals ($NO_3$) at night to form $SO_2$; and (c) $SO_2$ oxidation by OH in air and by $H_2O_2$ and
tropospheric ozone ($O_3$) in clouds (aqueous chemistry) to form sulfate (Chin et al 2000).
Henceforth, the skilful characterization of gas-phase radicals such as OH and $NO_3$ or compounds
like $O_3$ is essential for the representation of oxidation pathways in the atmosphere leading to the
formation of secondary aerosols (Jiménez et al 2003). Therefore, in this contribution the RACM
(Stockwell et al 1997, Geiger et al 2003) mechanism was coupled to GOCART through the kinetics
pre-processor (KPP) in WRF-Chem in order to provide the concentrations of radical and gas-phase
pollutants needed by the GOCART aerosol model. The Fast-J module (Wild et al 2000) was used as
photolysis option. Biogenic emissions were calculated using the Guenther scheme (Guenther et al
2006). Anthropogenic emissions coming from the Atmospheric Chemistry and Climate Model
Intercomparison Project (ACCMIP; Lamarque et al 2010) were kept unchanged in the simulation
periods (we considered the 2010 monthly values). Natural emissions depend on ambient conditions
and varied accordingly in our simulations following Ginoux et al 2001 for dust and Chin et al 2002
for sea salt.
The inclusion of aerosol-radiation interactions in the called ARI simulations follows Fast et al
(2006) and Chapman et al (2009). The overall refractive index for a given size bin was determined
by volume averaging associating each chemical constituent of aerosol with a complex index of
refraction. The Mie theory and the summation over all size bins were used to determine the
composite aerosol optical properties assuming wet particle diameters, taking into account the
humidity variations to allow variations of optical properties. Finally, aerosol optical properties were
transferred to the shortwave radiation scheme. Aerosol-cloud interactions were implemented by
linking the simulated cloud droplet number with the microphysics schemes (Chapman et al 2009)
affecting both the calculated droplet mean radius and the cloud optical depth. Although this WRF-
Chem version (3.6.1) does not allow a full coupling with aerosol-cloud interactions that includes the
aerosols exerting the highest influence from a climatic point point of view, i.e. sea salt and desert
dust, the microphysics implemented here is a modified version of  a single moment scheme that
turns it into a two-moment scheme in the simulations denoted ARCI. One-moment microphysical
schemes are unsuitable for assessing the aerosol-cloud interactions as they only predict the mass of
cloud droplets and do not represent the number or concentration of cloud droplets (Li et al 2008).
The prediction of two moments provides a more robust treatment of the particle size distributions,
which is key for computing the microphysical process rates and cloud/precipitation evolution. In
this sense, although the Lin microphysics is originally presented as a single moment scheme (Lin et
al 1983), a modified Lin double-moment microphysical scheme is implemented in WRF-Chem (Lin
et al 2008) and used here to conduct the ARCI simulations. In this scheme, both the mass and the
total number of cloud droplets are predicted. The prognostic treatment of cloud droplet number
involves water vapor, cloud water, rain, cloud ice, snow and graupel (Ghan et al 1997), and is
activated through the "mixactivate" module of WRF-Chem. In that module, WRF-Chem calculates
the aerosol number per volume concentration by using, for each aerosol type, the information about
the size (the mean volume-diameter of each aerosol mode, obtained from the aerosol mechanism
implemented in the simulation), and fixed densities and molecular weight of each type of aerosols.
With all this information and the total mass, WRF-Chem estimates the aerosol number for each
mode assuming spherical particles. The autoconversion of cloud droplets to rain droplets depends
on droplet number (Liu et al 2005). Droplet-number nucleation and (complete) evaporation rates
correspond to the aerosol activation and resuspension rates. Ice nuclei based on predicted
particulates are not treated. However, ice clouds are included via the prescribed ice nuclei
distribution, following the Lin et al (2008) scheme. Thus, the droplet number will affect both the
calculated droplet mean radius and cloud optical depth. Finally, the interactions of clouds and
incoming solar radiation were implemented by linking the simulated cloud droplet number with the
Goddard shortwave radiation scheme, representing the first indirect effect (i.e. increase in droplet
number associated with increases in aerosols), and with the Lin microphysics, representing the
second indirect effect (i.e. decrease in precipitation efficiency associated with increases in aerosols).
An important aspect of the differences in the model setup between experiments is that the
autoconversion scheme necessarily changes in the ARCI simulations as compared to the model
configuration used for ARI and BASE. The flag *progn* of the WRF namelist should be set to 0 for
running ARI experiments in order to keep disabled the interaction of the online-estimated aerosols
with cloud microphysics, hence ensuring the use of prescribed aerosols (as in the case of the BASE
simulations) as this regards. Conversely, *progn* should be set to 1 for running ARCI experiments in
order to feed the cloud microphysics scheme with the online-estimated number and physico-
chemical properties of aerosols (this effectively turns the Lin scheme into a second-moment
microphysical scheme).
**2.3 – Data and methods**
The WRF and WRF-Chem outputs were recorded every hour for surface downward solar radiation
(RSDS), total cloud cover (CCT) and the concentrations of various aerosol species (dust, black
carbon, organic carbon and sea salt). The concentration of sulfates was indirectly computed from
the recorded concentrations of $SO_2$ and OH using the same kinetic reaction implemented in the
RACM-KPP module. From the concentrations of the five aerosol species, the atmospheric optical
depth (AOD) at 550 nm was estimated using the reconstructed mass-extinction method (Malm et al
1994), as in Palacios-Peña et al (2020). The RSDS and CCT data simulated by the driving GCM
runs were used for comparison purposes. We also retrieved the AOD at 550 nm as seen by the GCM
from the MACv2 data (Kinne et al 2019), whose anthropogenic changes are in accordance with the
RCP8.5 while its coarse mode (of natural origin) was not allowed to change. Also, RSDS values
from the ERA5 reanalysis (Hersbach et al 2020) were used for validation purposes. Seasonal means
means of all the variables were used in the analysis. These means involve all the records within
each season in the series.
We also studied the sensitivity to resolving aerosol interactions of RSDS and AOD under clear-sky
conditions. The analysis in absence of cloudiness will tell us more about the relevance of the direct
radiative effect of aerosols. RSDS and AOD clear-sky ($RSDS_{cs}$ and $AOD_{cs}$, respectively) mean
seasonal series were constructed as follows. First, hourly series of CCT, RSDS and AOD were time
averaged up to the daily timescale. Second, days with CCT values lower than 1% were retained
(this criterion is applied at the grid-box level, for each grid-box individually); otherwise we put a
missing value. These clear-sky daily series were then time averaged up to the seasonal time-scale.
When pairs of experiments were compared, only coincident clear-sky dates (days) in the series were
selected (missing values were also assigned in this case to the non-coincident dates with clear-sky
conditions) before performing the seasonal time average. This resctriction aims to avoid the
masking effect of Earth orbit related issues, of large scale climate drivers and/or local forcings such
as water vapor content, since different days may have different daytime lengths and different
atmospheric compositions (different atmospheric optical depth or atmospheric transmissivity) that
may mask the AOD effect under clear-sky conditions. The analysis involving $RSDS_{cs}$ and $AOD_{cs}$
was carried out only over those grid points where at least 75% of the summer mean values in the
series (i.e. at least 15 records per period) were not missing (which, according to our methodology,
would occur only if all days within a summer season had CTT values $\geq 1\%$).
Spatial correlations between climatological patterns were computed excluding sea grid points,
considering absolute values in case they involved differences (while these were depicted in the
Figures in relative terms, i.e. in %), using the CDO *fldcor* function
(https://code.mpimet.mpg.de/projects/cdo/embedded/cdo.pdf). Temporal correlations were
computed at the grid point level between the seasonal series, considering absolute values in case
they involved differences, using the R *cor* function

(https://www.rdocumentation.org/packages/stats/versions/3.6.2/topics/cor; *Pearson* correlation coefficient selected). The statistical significance of any signal was assessed with a t-test imposing $p<0.05$.

We focus on the summer season (JJA), when solar energy is at its maximum, AOD typically reaches high values and the aerosol radiative effect has been proven to be strongest (Pavlidis et al 2020).

In order to investigate the underlying mechanisms explaining the signals found in RSDS and CCT, additional variables and statistics were used, namely: JJA-mean top-of-the-atmosphere outgoing short-wave radiation (RSOT), surface (2 m height) air temperature (TAS), surface (1000 hPa pressure level) relative humidity (RH), total precipitation (PR) and convective precipitation (PRC); number of cloudy days (CLD, defined as days with mean CCT>75%) in the summer series; $90^{th}$ percentile of the JJA day-mean PR series; and number of rainy days (RD, defined as days with mean precipitation > 1 mm) in the JJA daily PR series. Vertical profiles of air temperature (T) and cloud fraction (CLFR) were also considered.

## 3 – Results

### 3.1 – Historical patterns

*Brief validation of the simulated RSDS patterns*

As a first test, Supp Fig 2 provides the GCM, ERA5 BASE, ARI and ARCI JJA climatologies of RSDS in the historical period and the results of a brief validation exercise. Although the five patterns depict similar structures (Supp Fig 2a,b,d-f), Supp Fig 2g-i reveals significant deviations of the climatologies from the WRF experiments with respect to the GCM: positive values (higher RSDS values in the RCM experiments) south and northward (up to 20 and 30% respectively), and negative values in between (10-15%, eventually up to 25%). These differences are very similar to those obtained when WRF climatologies are compared with the ERA5 pattern (Supp Fig 2j-l), with a notable exception over the Scandinavian region where the agreement between the WRF experiments and ERA5 is higher than between the WRF experiments and the GCM. In fact, the GCM pattern strongly underestimates RSDS over such a region (over 30%; Supp Fig 2c), while showing a better agreement with ERA5 elsewhere as compared to the WRF simulations.

*Aerosols impact on the simulated RSDS patterns*
Although the three WRF experiments (BASE, ARI and ARCI) perform similarly when compared to
the GCM or ERA5, there are still noticeable differences between them (Fig 1a-c and Supp Fig 3a-
c), and it is there that this research focuses. The inclusion of aerosols (ARI and ARCI experiments)
reduces the JJA mean values of RSDS in central and northern parts of our domain by a few
percentage points (i.e. by ~10 $Wm^{-2}$) as compared to the BASE experiment (Fig 1a,b and Supp Fig
3a-b). This reduction is generally stronger in ARI than in ARCI. Consequently, the ARCI minus
ARI pattern (Fig 1c and Supp Fig 3c) depicts mostly positive values (by ~5 $Wm^{-2}$) over central and
southern regions. This result indicates that the indirect aerosols effects tend to counteract the joint
direct and semi-direct effects seen in the ARI minus BASE pattern, reducing it by up to a half over
most of the domain, which is in agreement with previously reported findings (Pavlidis et al 2020).
In order to better understand the patterns of differences in RSDS between experiments, Fig 1 (and
Supp Fig 3) also provides differences in CCT and AOD (panels d to f and g to i, respectively) and
the spatial correlations (*s_corr*) between these patterns and those of RSDS differences.
*The role of CCT*
Compared to BASE, both ARI and ARCI lead to more cloudiness in central and northern regions
(albeit quite slight increases, well below 5%). This could respond to the direct effect of the
scattering of the solar radiation due to the high presence of sea salt, dust and sulfate over these areas
(Fig 2), as an increase in RSOT over these areas is also appreciated in both ARI and ARCI
simulations (Fig 3a-b). In addition, this direct effect could be triggering the following feedback
mechanism: the cooling effect downward (where less solar radiation is received because of its
scattering) cools down surface temperatures (Fig 3d-e), thus increasing relative humidity (Fig 3g-h),
which may favor the formation of  clouds (these should be non-convective, mostly low-level, clouds
as the decrease in TAS leads to more stable atmospheric layers; Fig 4a,b), thus less radiation
reaches the surface, thus lower surface temperatures, and so on. Noteworthy, both the reduction in
RSDS and the accompanying increase in RSOT is more marked in ARI than in ARCI over central
regions (Fig 1c and Fig 3c), where the indirect effects included in the ARCI simulation, such as in-
cloud aerosol scavenging processes, could lead to cleaner atmospheres than ARI simulates.
Conversely, both ARI and ARCI lead to less cloudiness southward as compared to BASE, especially
ARCI (reductions up to 10% in Mediterranean regions; Fig 1d-e). Consistently, the ARCI minus
ARI pattern (Fig 1f) depicts negative values (around 5%) along the Mediterranean strip. Therefore,
both semi-direct and indirect aerosol effects would tend to diminish cloudiness southward, with the
latter (indirect effect) having the greatest impact. This could be due to the fact that a high presence
of large aerosols over southern Europe, both in form of dust or sulfate in our case (Fig 2), can
accelerate collision-coalescence processes fastening that precipitation occurs and thus shortening
the lifetime of clouds (Lee et al 2008), which is most plausible in the warm season over warm areas
(Yin et al 2000), as long as aerosol-cloud interactions are resolved by the model. However, we did
not find such an enhanced precipitation effect in our simulations (maybe the signal does not hold at
the climatic scales assessed here), only a decrease in both mean cloudiness and number of cloudy
days (Supp Fig 3j-l) together with consistent pictures of lower mean precipitation, lower mean
convective precipitation, fewer rainy days and lower extreme precipitation values emerging over
those areas where the aerosol effects diminish cloudiness (Fig 5). The reduction in convective
precipitation (the prevailing form of precipitation over this area during the summer season) suggests
that absorption might be creating more stable atmospheric situations (by heating aloft layers) and
thus preventing clouds formation via convective phenomena and increasing the incoming surface
solar radiation. But we did not find any clear evidence of that either (Fig 4c). So the thermodynamic
effect of aerosols on clouds inhibition and burn-off might justify the reduction in CCT (mainly at
low levels; Fig 4d) and the accompanying increase in RSDS in the southernmost areas. These
signals are intensified when we add the indirect aerosols effects, likely due to the removal of
aerosols via scavenging processes, which cleans the atmosphere favoring that the solar radiation
reaches the surface.
Whatever the underlying mechanisms are, the patterns of differences between experiments in CCT
are well correlated with the corresponding patterns of differences in RSDS, thus indicating a key
role of CCT in driving the latter. Indeed, the temporal correlation at the grid point level between the
seasonal series of RSDS and CCT differences is above 0.8 (negative) in most of the domain (Supp
Fig 4a-c).
***The role of AOD***
The inclusion of aerosols also leads to differences of a few percentage points (2-5%) in the AOD
values between ARCI and ARI simulations over western areas (Fig 1i), and the AOD climatologies
from these two experiments provide a consistently non-null picture (Fig 1g,h; null values can be
considered for BASE). However, the patterns for AOD do not correlate with those for RSDS and
the seasonal series of differences in AOD hardly correlates with the seasonal series of differences in
RSDS except for certain locations of central and southeast Europe (Supp Fig 4d-f). Interestingly,
over these locations, the temporal correlation between differences in RSDS and differences in AOD
are positive, indicating the secondary role of the direct radiative effect of the aerosols there: if the
larger the AOD, the larger the RSDS, it is because semi-direct and indirect effects counteract the
impact of the direct scattering effect.

*Clear-sky analysis*

An overall predominant link between the aerosol effect on cloudiness and its impact on the amount
of solar radiation reaching the surface, that totally masks any other mechanism related to the
variation in AOD and its direct impact on RSDS, has been detected so far. On the contrary, as
expected, under clear-sky conditions, both the negative spatial correlations between the patterns of
$AOD_{cs}$ and $RSDS_{cs}$ differences between experiments (Fig 6), and the negative temporal correlations
between the respective series computed at the grid point level (Supp Fig 4g-i), support the relevant
role of the $AOD_{cs}$ variable for the simulation of $RSDS_{cs}$. The differences in $RSDS_{cs}$ between ARI or
ARCI and BASE are negative (around 5 $Wm^{-2}$; Fig 6 and Supp Fig 5) over the study area (restricted
to the southern half of the domain since the clear-sky series northward lack of sufficient records to
perform a robust statistical analysis), illustrating the direct radiative effect of aerosols and further
supporting the important role of semi-direct and indirect effects (that make the negative clear-sky
signals softer and even positive over some southern locations, as shown in Fig 1a,b). ARCI minus
ARI differences in $RSDS_{cs}$ are basically null since semi-direct and indirect effects are largely
irrelevant in the absence of cloudiness.

**3.2 – Future projections**

*Future climatologies*

The overall results described above also hold under future climate conditions, while some
differences were identified and deserve mention. The inclusion of aerosols reduces RSDS over most
of the domain due to direct, semi-direct and indirect effects (Supp Fig 6a-c). In particular, this
occurs significantly southward, along the Mediterranean strip, in contrast to the previous results.
Over some locations, mainly in central Europe, this reduction is stronger in ARI than in ARCI, as
detected under historical conditions. However, the opposite (larger RSDS reduction in ARCI than in
ARI) occurs elsewhere, interestingly over the Mediterranean strip, which also contrasts with the
results found under historical conditions. These results further support the sensitivity of the
simulations to both aerosol-radiation and aerosol-cloud interactions under changed climates, in such
a way that cloudiness still appears to be the most important explanatory variable for the differences
in RSDS between experiments, although the role of AOD gains much relevance as compared to the
analysis under historical conditions (see the spatial and temporal correlation values in Supp Fig 6d-i
and Supp Fig 7a-f, respectively). Under clear-sky conditions (Supp Fig 7g-i and Supp Fig 8), the
results are identical to those reported in the previous section.
Therefore, what contrasts most with the previous results is that (1) both ARI and ARCI simulations
provide diminished values of RSDS (of a few percentage points but statistically significant) over
southern locations as compared to BASE (Supp Fig 6a,b), which should primarily respond to the
direct aerosol effect of scattering the radiation (enhanced RSOT can be appreciated in Supp Fig
9a,b) since it occurs, in particular, in spite of the diminished CCT values simulated by the ARI
experiment there (Supp Fig 6d); and (2) such a reduction in RSDS over such southern locations is
reinforced when indirect effects are included (Supp Fig 6c), as these do cause higher CCT values
than BASE (Supp Fig 6e) and, consequently, higher RSOT values there than ARI (Supp Fig 9a-c).
This latter could also respond to the added role of aerosols in modifying the optical properties of
clouds. When ACI are considered, aerosols act as cloud condensation nuclei, which can lead to
whiter clouds with higher albedo. Interestingly, but out of the scope of this study, different PR shifts
east and west across Mediterranean Europe were detected when ARCI and ARI experiments were
compared between them, and then ARCI and ARI with BASE (Supp Fig 10). Over the Balkan
Peninsula (south-east of the domain), ACI enhances precipitation, whether in the form of convective
precipitation, total precipitation, intense precipitation or number of rainy days, more than ARI does,
whereas over the Iberian Peninsula (south-west of the domain), ARI leads to higher precipitation
rates and intensity, while reducing the frequency of rainy days as compared to ARCI. These signals
suggest that the fact that different aerosol species prevail in these areas (the concentration of sulfate
is larger eastward, while the concentration of dust particles is larger westward; Supp Fig 11), and
how this affects the ratio between large and fine particulate matter, might have an impact along with
the aforementioned  mechanisms in this case (López-Romero et al 2020).
Since the patterns of differences in the analyzed variables show different structures under historical
and future climate conditions, the RSDS change patterns vary when ARI and ACI are taken into
account by the model, as described below.
*Future projections*
The change patterns for RSDS are similar in both the BASE and ARI experiments (Fig 7b,c  and
Supp Fig 12b,c), showing negative signals in northernmost regions (up to 10%, ~15 Wm$^{-2}$) and
positive signals southward (up to 5%, again ~15 Wm$^{-2}$). The latter are more widespread in ARI than
in BASE, which makes the ARI pattern the most similar to the change pattern from the GCM (Fig
7a and Supp Fig 12a). However, when aerosols-cloud interactions are included in the WRF runs,
such a positive RSDS change signals mostly disappear, while the northern negative ones reinforce
in some parts as compared to the ARI pattern (Fig 7d and Supp Fig 12d). These results are in quite
good agreement with the corresponding change patterns for CCT (Fig 7e-h and Supp Fig 12e-h) –
including the fact that the negative change signals for CCT appearing southward in the GCM,
BASE and ARI experiments are much less evident in ARCI – and occur in spite of two constraining
facts regarding the AOD simulation approach in our WRF experiments: (1) AOD remains
unchanged in the BASE experiment (as illustrated by Fig 7j), and (2) AOD changes from the ARI
and ARCI experiments are hardly realistic because their anthropogenic component is disregarded
(as specified in Section 2), and thus depict patterns (Fig 7k,l) that have nothing to do with the GCM
projection in Fig 7i (which does consider time evolving anthropogenic aerosols). In fact, the spatial
correlation between the patterns of AOD and RSDS changes is lower than between those of CTT
and RSDS changes. Therefore, direct and semi-direct aerosol effects have a limited impact on the
RSDS future projections here, while indirect effects play a major role by reducing the future
decrease in CCT southward within our domain and thereby dispelling the future increase in RSDS
in this region.
The change signals for RSDS$_{cs}$ and AOD$_{cs}$ (Fig 8 and Supp Fig 13) depict different spatial structures
to those for RSDS and AOD, turning mostly negative southward and positive northward for RSDS$_{cs}$
(with negative signals around 5% and positive up to 10%, in both cases implying changes up to 20
Wm$^{-2}$). Although this occurs similarly in the three experiments (BASE, ARI and ARCI), BASE
provides the softest signals, which does evince a certain role of the direct aerosol effect. However,
there is not a clear relationship between AOD$_{cs}$ change patterns and RSDS$_{cs}$ changes (low spatial
correlation), except for some local signals in the north-east where the direct aerosol effect enhances
RSDS$_{cs}$ in areas with reduced AOD$_{cs}$. However, as discussed above, the role of retaining, or not,
coincident clear-sky dates between pairs of experiments is important in filtering out the true role of
AOD$_{cs}$ on RSDS$_{cs}$. Thus, the fact that change patterns are constructed over different dates could
partially explain the apparently negligible role of AOD$_{cs}$ on RSDS$_{cs}$ in this case. But only partially,
as the BASE change pattern for RSDS$_{cs}$  (simulated on the ground of nule AOD$_{cs}$ changes)
resembles the respective patterns from ARI and ARCI experiments.
**4 - Discussion and conclusions**
We presented here a research on the role of dynamically modeled atmospheric aerosols in regional
climate simulations with a focus on the impacts on the solar resource during the summer season
from a climatic perspective, including projected changes to a medium-range horizon and analysis
under clear-sky conditions. For this, we evaluated a set of 20-yr long runs (spanning both historical
and future periods) without including aerosols and with resolved aerosol-radiation and aerosol-
radiation-cloud interactive (two-way) interactions performed with the WRF model (BASE, ARI and
ARCI experiments, respectively).
We interpreted the signals on the basis that the differences between ARI and BASE can be
attributed to direct and semi-direct aerosol effects and the differences between ARCI and ACI to the
indirect aerosol effect. Nonetheless, we should acknowledge that the autoconversion scheme called
so that cloud droplets can turn into rain droplets in the ARCI simulations is different to the
autoconversion scheme activated in the ARI (and BASE) simulations. This change in the WRF-
Chem configuration can lead to differences between ARCI and ARI experiments that do not come
necessarily from the aerosol-cloud interactions from a physical point of view (Liu et al 2005). In
fact, the activation of the aerosol-cloud interactions requires further changes in the model
configuration (as compared to the configuration used for the simulations labeled ARI) beyond the
autoconversion scheme, such as the activation of aqueous chemistry processes, which could also
have an added impact to the effect that can be strictly attributed to the aerosol-cloud interactions.
However, technically, the encoding of the WRF-Chem model hampers better isolation of the effect
of the aerosol-cloud interactions (the mentioned aspects necessarily change between ARI and ARCI
run modes). Therefore, ARCI-ARI differences can not be attributed to the aerosol-cloud interactions
from a purely physical point of view, but to the activation of the aerosol-cloud interactions from a
modeling point of view. It should also be borne in mind that the set of experiments performed
allows any attribution to the interactive aerosol modeling approach adopted here to be made, while
it is a distinct feature with respect to previous studies aimed at providing more consistent signals
from a physical point of view. Besides, and more general, internal variability plays a role in the
simulations (e.g. Gómez-Navarro et al 2012), and a single member with a single physics
configuration, as was used for the sensitivity experiment, may not be sufficient to obtain generally
occurring responses. Last, we kept the anthropogenic aerosol emissions unchanged throughout the
simulation period. This approach permits to better isolate the signals from the aerosol-radiation-
cloud interactions due to the climate variability alone and the so-called climate change penalty
alone, but at the expense of the reliability of the simulated patterns. Anthropogenic emissions have
been dramatically reduced since the 1980s and are expected to continue in that pathway to the
future (IPCC 2013, 2014), so keeping 2010 values (as we did) could lead to an underestimation of
AOD in the historical period (in fact, it does; reference AOD climatologies can be found in Pavlidis
et al. 2020) and to its overestimation in the future period. Under these constraints, we draw the
following conclusions.
The inclusion of aerosols in the WRF simulations reduces in general the amount of solar radiation
reaching the surface by a few percentage points (~5%) under both historical and future climate
scenarios, as expected (Nabat et al 2015a, Gutiérrez et al 2018, Pavlidis et al 2020). Under historical
conditions, this effect is larger when the aerosol-cloud interaction remains turned off, because its
activation leads to less cloudiness (over Mediterranean Europe) and lower AOD values (over
Atlantic Europe), as evidenced when ARCI and ARI simulations were compared. The differences in
RSDS between experiments are in overall good agreement with those found in cloudiness, while
they seem to be unlinked with the differences in AOD in many parts of the domain. In agreement
with Pavlidis et al (2020), AOD plays its major role under clear-sky conditions. However, the
differences in JJA-mean values of RSDS under clear skies between experiments with and without
dynamic aerosols are hardly about 1%, while still significant in some of the southernmost parts of
our European domain, and almost null between ARCI and ARI.
Our results suggested a variety of drivers underlying the mechanisms to explain the signals
obtained, depending on the region (and season; winter plots are provided in Supp Fig 14-17 as an
example for interested readers), and varying under future climate conditions. These involve the
scattering of solar radiation with the consequent cooling downward, suppression of cloudiness due
to thermodynamic effects, modification of the clouds' optical properties, or in-cloud scavenging
processes. As these prevailing mechanisms change (up to a point) in the future, the sensitivity of the
WRF simulations under future climate conditions, represented through the patterns of differences in

RSDS, is somehow depicted differently than under historical conditions. Therefore, the future projections also show sensitivity to the way the model considers aerosols.

The patterns of change for RSDS and CCT again show high spatial correlations in all the GCM and RCM (BASE, ARI and ARCI) projections. Although lower, still high spatial correlations define the match between the RSDS change patterns and those for AOD in the GCM, while this is not the case in either the ARI or ARCI experiments. The GCM, BASE and ARI experiments agree in projecting positive RSDS change signals in southern and eastern areas (around 5%), while clear differences are found between the GCM and the BASE or ARI RSDS change patterns (with the latter two very similar) in central and northeastern areas, where the positive signals from the GCM turn notably negative in both BASE and ARI. ARCI provides the most singular and negative picture of RSDS changes among all those shown, with widespread decreasing signals of a few percentage points, further reinforcing the fact that the indirect effect tends to counteract the direct and semi-direct effect of aerosols and enlarges the distance between the RCM and the GCM projections.

Previous works (Jerez et al 2015, Sørland et al 2018) had already detected inconsistencies in the change signals between RCM projections and those from their driving GCM, which have been related to the way aerosols had been represented in the RCM through their impact on the simulated AOD (Gutiérrez et al 2020, Boé et al 2020), and in particular to their direct and semi-direct effects and their reduced concentrations in the future as long as anthropogenic emissions are projected to decrease. In agreement with these previous findings, insofar as we kept the anthropogenic aerosol emissions unchanged throughout the simulation period, our projections differ from those obtained with the GCM. Nevertheless, the ARI experiment brings our results slightly nearer to those of the GCM as compared to the BASE experiment, perhaps also indicating the key role of the direct and semi-direct aerosol effects for reducing the GCM-RCM discrepancies, as reported in these previous works. However, pushing our understanding further, by turning off the already reported effect of reduced aerosol concentrations in the future via the direct and semi-direct effects, our approach made it possible to identify the prevailing role of CCT changes (over the dynamically simulated natural changes in AOD) to explain our signals of change in RSDS, and the capacity of the aerosol-radiation-cloud interactions to significantly alter our RSDS change patterns (much more than aerosol-radiation interactions alone do). Thus, although change patterns for RSDS certainly look uniform among experiments under clear-sky conditions (likely because we suppressed the anthropogenic component for the changes in AOD, which was identified by Boé at al (2020) as a main factor for these signals indeed), the results presented here further indicate that the joint effect

of aerosol-radiation-cloud interactions should be considered in the RCM simulations that serve to build up action-oriented messages in the challenging context of current climate change, calling for caution otherwise and for future research efforts in this line.

## Author contribution

S. J. conceived this study. L. P.-P., P. J.-G. and J. P. M. designed the experiments and J. M. L.-R. and E. Pravia-Sarabia carried them out. S. J. performed the analysis and prepared the manuscript with contributions from all co-authors.

## Acknowledgments

This study was supported by the Spanish Ministry of Science, Innovation and Universities/*Agencia Estatal de Investigación* and the European Regional Development Fund through the projects EASE (RTI2018-100870-A-I00, MCI/AEI/FEDER, UE) and ACEX (CGL2017-87921-R, MINECO/AEI/FEDER, UE), and by the *Fundación Séneca – Agencia de Ciencia y Tecnología de la Región de Murcia* through the project CLIMAX (20642/JLI/18). S. Jerez receive funding from the *Plan Propio de Investigación* of the University of Murcia (grant No. UMU-2017-10604). L. Palacios-Peña thanks the FPU14/05505 scholarship and the Spanish Ministry of Education, Culture and Sports. J. M. López-Romero acknowledges the FPI-BES-2015-074062 grant from the Spanish Ministry of Science.

## Code and data availability

All data (netcdf files) and relevant codes (scripts for data analysis and WRF namelists) for reproducing this study are publicly accessible at http://doi.org/10.23728/b2share.a65d25c2b3ba49e1a46e970783e9476e.

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

 **Figure caption**

 **Figure 1.** Relative differences between the WRF simulations in the RSDS (a to c), CCT (d to f) and
 AOD at 550 nm (g to i) summer (JJA) climatologies in the historical period (1991-2010), squared if
 statistically significant (p<0.05); units: %. Note that panels g and h are referred to the horizontal
 colorbar just below them and simply represent the AOD summer climatologies in ARI and ARCI
 respectively. Spatial correlations (*s_corr*) between the patterns in the second and third rows and the
 respective patterns in the first row are indicated in the headers.

 **Figure 2.** Contribution of each aerosol species (BC: black carbon, DUST, OC: organic carbon,
 SEAS: sea salt, and SULF: sulfate) to the JJA-mean total surface aerosol mass concentration in ARI
 and ARCI simulations in the period 1991-2010. Units: %.

 **Figure 3.** Relative differences between the WRF simulations in the top-of-the-atmosphere outgoing
 short-wave radiation (RSOT, a to c), surface (2 m height) air temperature (TAS, d to f), surface
 (1000 hPa pressure level) relative humidity (RH, g to I), and number of cloudy days (CLD; defined
 as days with mean CCT>75%, j to l) summer (JJA) climatologies in the historical period (1991-
 2010), squared if statistically significant (p<0.05). Units: K for TAS, % for RSOT, RH and CLD.

 **Figure 4.** Vertical profiles of the spatial mean differences in summer (JJA) mean air temperature (T,
 left panels) and cloud fraction (CLFR, right panels) in the historical period (1991-2010) between
 experiments over two small areas: a northern one (Region N; top panels) and a southern one
 (Region S; bottom panels), gray shaded in the respective maps. These are plain differences, which
 units are K for T and % for CLFR.

 **Figure 5.** Relative differences between the WRF simulations in the summer (JJA) climatologies of
 various precipitation (PR) statistics in the historical period (1991-2010), squared if statistically
 significant (p<0.05): mean PR (a to c), 90[th] percentile of the JJA daily PR series (d to f), number of
 rainy days (RD) in the JJA daily PR series (defined as days with mean precipitation >1 mm, g to I),
 and mean convective precipitation (PRC, j to l). Units: %.

 **Figure 6.** Relative differences between the WRF simulations in the $RSDS_{cs}$ (a to c) and $AOD_{cs}$ at
 550 nm (d to f) summer (JJA) climatologies, this is under clear-sky conditions, in the historical
 period (1991-2010), squared if statistically significant (p<0.05); units: %. Note that panels d and e

are referred to the horizontal colorbar just below them and simply represent the AOD summer
climatologies in ARI and ARCI, respectively. Gray shaded areas depict grid points where less than
75% of the summer mean values in the time series of $RSDS_{cs}$ and $AOD_{cs}$ were not missing. Spatial
correlations (*s_corr*) between the patterns in the second row and the respective patterns in the first
row are indicated in the headers.
**Figure 7.** Projected changes for the RSDS (a to d), CCT (e to h) and AOD at 550nm (i to l) summer
(JJA) climatologies by the GCM (first column) and the WRF experiments (second to fourth
columns); units: %. Squares highlight statistically significant signals ($p<0.05$). Note that panel i is
referred to the horizontal colorbar just below it. Spatial correlations (*s_corr*) between the patterns in
the second and third rows and the respective patterns in the first row are indicated in the headers.
**Figure 8.** Projected changes for the $RSDS_{cs}$ (a to c) and $AOD_{cs}$ at 550nm (d to f) summer (JJA)
climatologies, this is under clear-sky conditions, by the WRF experiments, squared if statistically
significant ($p<0.05$); units: %. Gray shaded areas depict grid points where less than 75% of the
summer mean values in the time series of $RSDS_{cs}$ and $AOD_{cs}$ were not missing  in either the
historical or the future period. Spatial correlations (*s_corr*) between the patterns in the second row
and the respective patterns in the first row are indicated in the headers.

# RSDS, CCT & AOD JJA climatologies for 1991-2010: differences between experiments

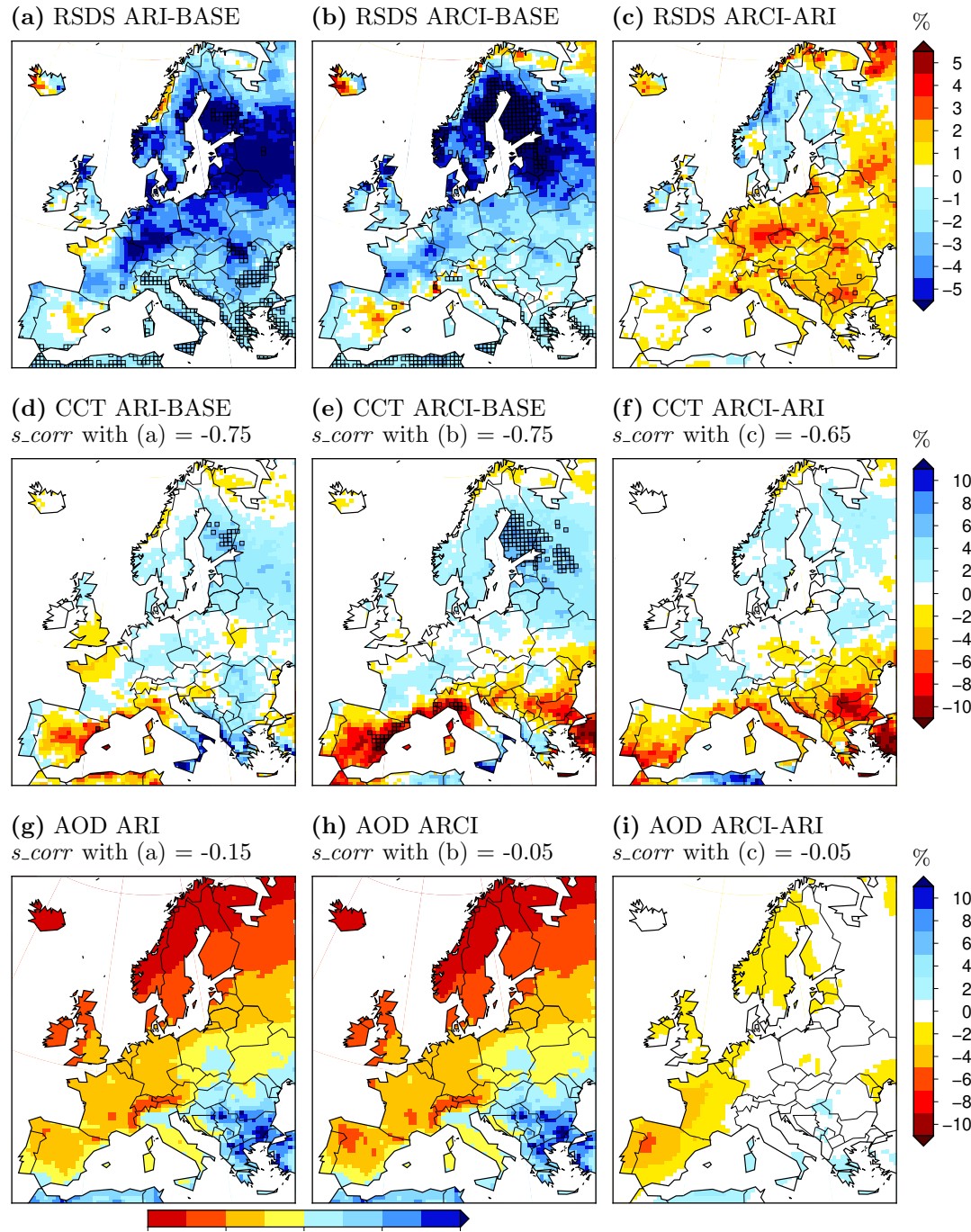

**(a)** RSDS ARI-BASE

**(b)** RSDS ARCI-BASE

**(c)** RSDS ARCI-ARI

**(d)** CCT ARI-BASE
*s_corr* with (a) = -0.75

**(e)** CCT ARCI-BASE
*s_corr* with (b) = -0.75

**(f)** CCT ARCI-ARI
*s_corr* with (c) = -0.65

**(g)** AOD ARI
*s_corr* with (a) = -0.15

**(h)** AOD ARCI
*s_corr* with (b) = -0.05

**(i)** AOD ARCI-ARI
*s_corr* with (c) = -0.05

Figure 1

**Contribution of each aerosol species to the JJA-mean total surface aerosol concentration (period 1991-2010)**

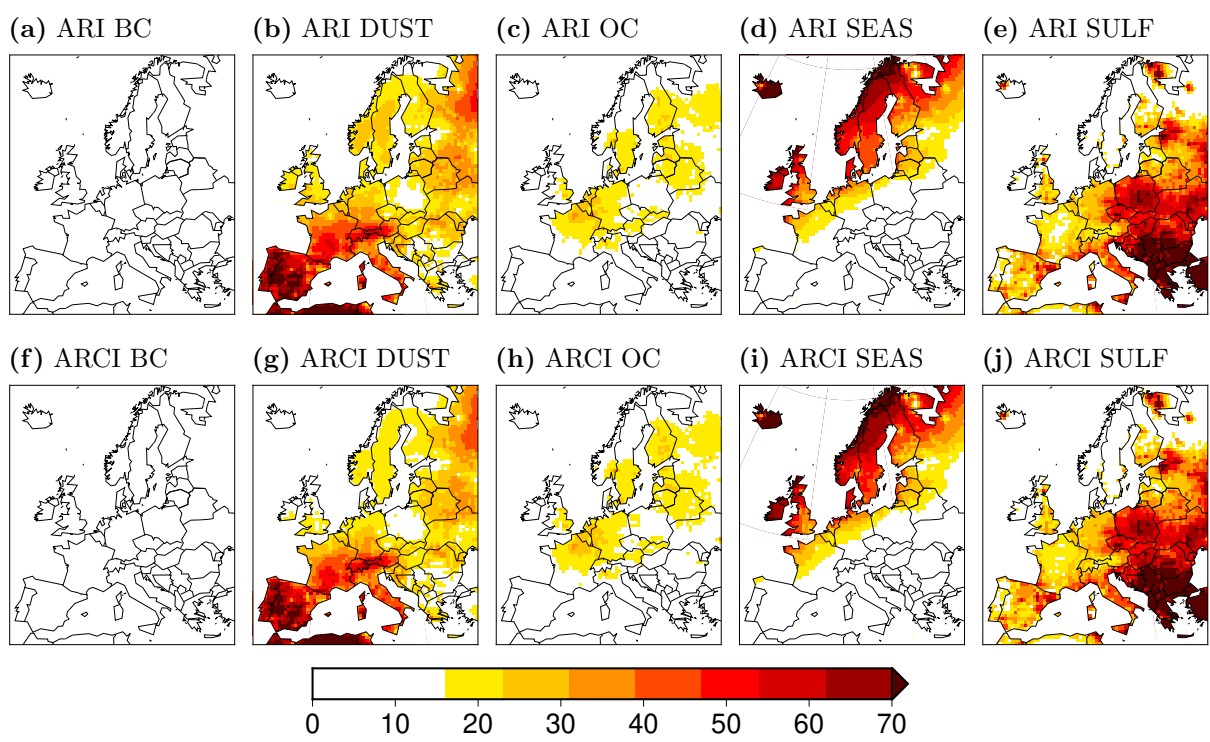

Figure 2

## RSOT, TAS, RH & CLD JJA climatologies for 1991-2010: differences between experiments

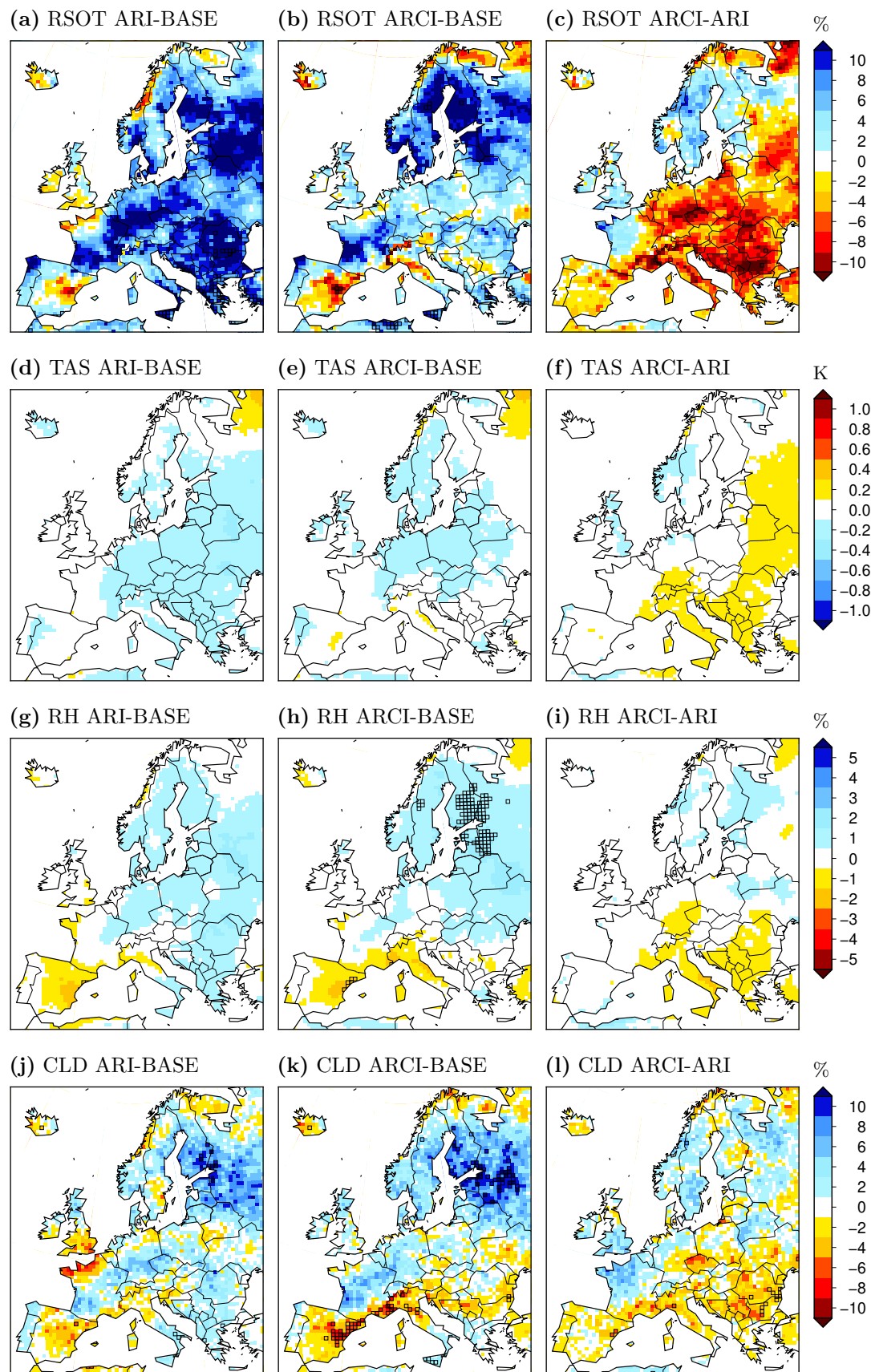

**Figure 3**

## Vertical profiles of differences in JJA-mean T and CLD in the period 1991-2010

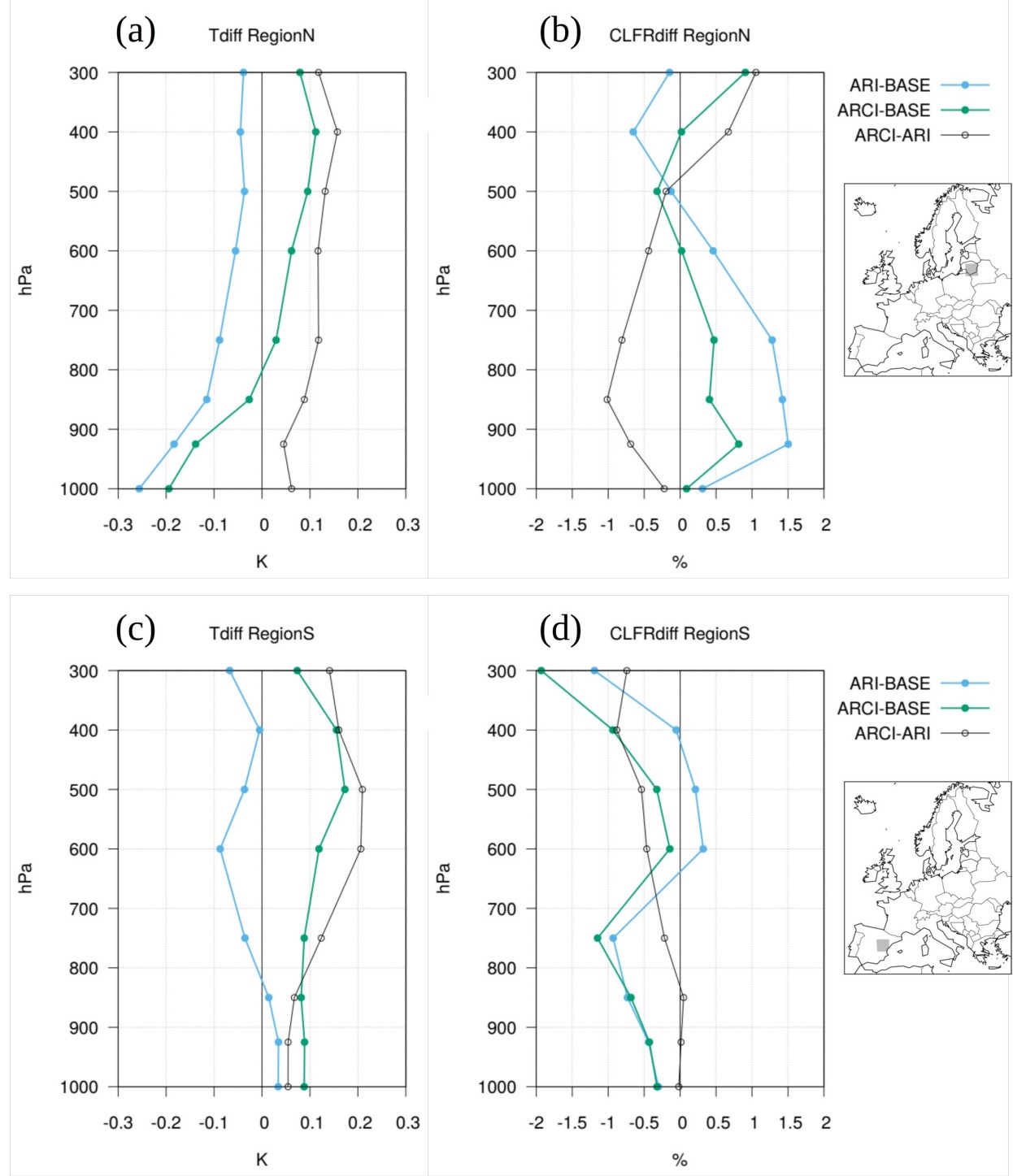

Figure 4

**Precipitation-related JJA climatologies for 1991-2010: differences between experiments**

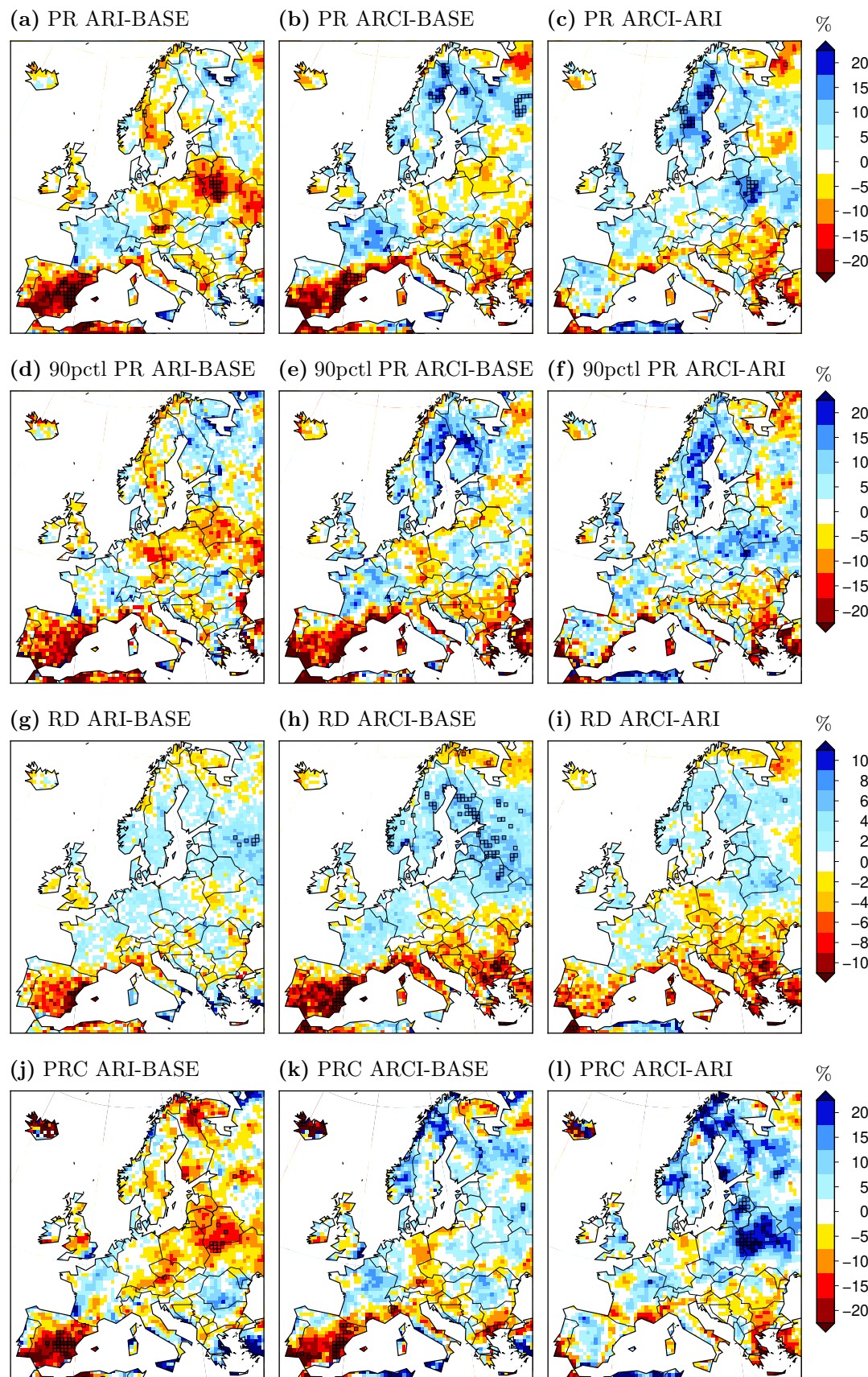

**Figure 5**

**RSDS$_{cs}$ & AOD$_{cs}$ JJA climatologies for 1991-2010:**
**differences between experiments**

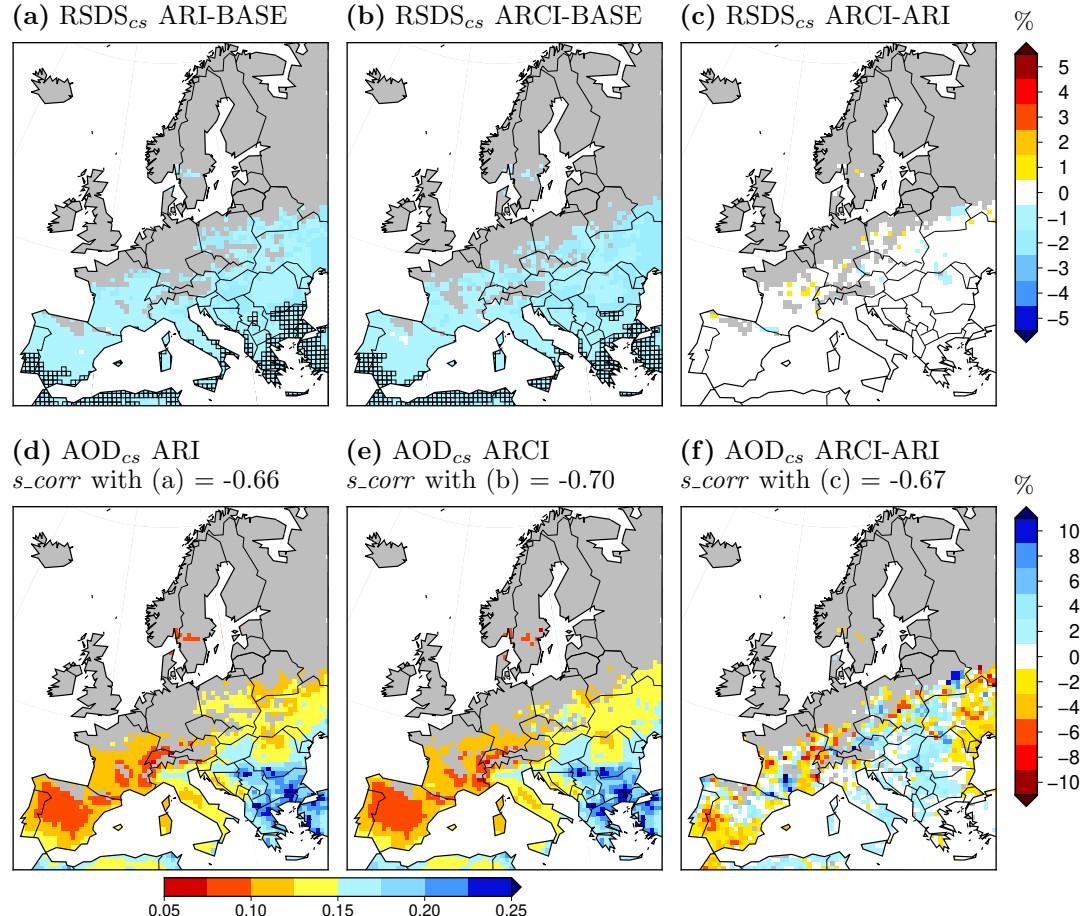

**Figure 6**

# RSDS, CCT & AOD JJA changes (2031-2050 vs. 1991-2010)

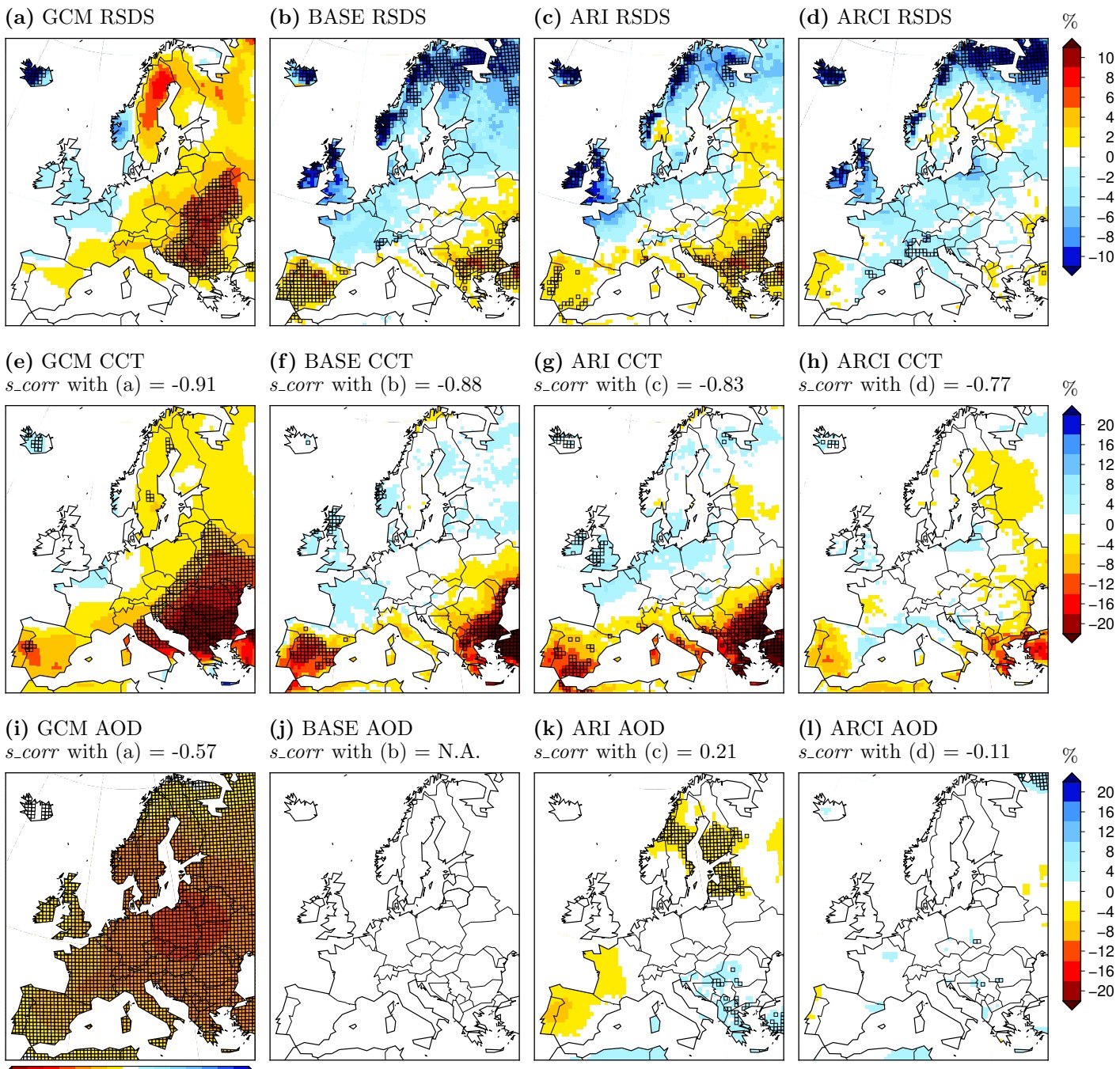

**(a)** GCM RSDS

**(b)** BASE RSDS

**(c)** ARI RSDS

**(d)** ARCI RSDS

**(e)** GCM CCT
*s_corr* with (a) = -0.91

**(f)** BASE CCT
*s_corr* with (b) = -0.88

**(g)** ARI CCT
*s_corr* with (c) = -0.83

**(h)** ARCI CCT
*s_corr* with (d) = -0.77

**(i)** GCM AOD
*s_corr* with (a) = -0.57

**(j)** BASE AOD
*s_corr* with (b) = N.A.

**(k)** ARI AOD
*s_corr* with (c) = 0.21

**(l)** ARCI AOD
*s_corr* with (d) = -0.11

**Figure 7**

**RSDS$_{cs}$ & AOD$_{cs}$ JJA changes (2031-2050 vs. 1991-2010)**

**(a)** BASE RSDS$_{cs}$      **(b)** ARI RSDS$_{cs}$      **(c)** ARCI RSDS$_{cs}$

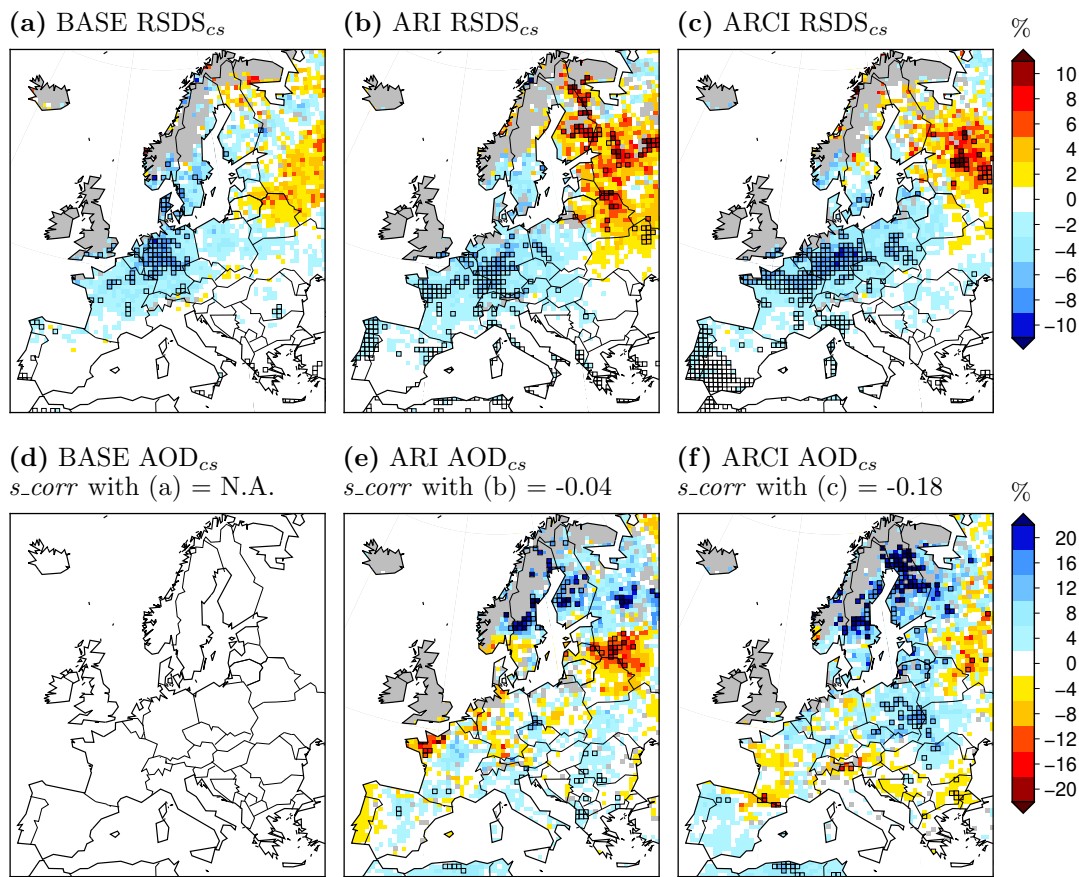

**(d)** BASE AOD$_{cs}$
*s_corr* with (a) = N.A.

**(e)** ARI AOD$_{cs}$
*s_corr* with (b) = -0.04

**(f)** ARCI AOD$_{cs}$
*s_corr* with (c) = -0.18

**Figure 8**