# Peer review of "Sensitivity of surface solar radiation to aerosol-radiation and aerosol-cloud interactions over Europe in WRFv3.6.1 climatic runs with fully interactive aerosols"

_Geoscientific Model Development, 2020_

## Short Comment (SC1) · 12 Aug 2020

Dear authors,

in my role as Executive editor of GMD, I would like to bring to your attention our Editorial version 1.2:

https://www.geosci-model-dev.net/12/2215/2019/

This highlights some requirements of papers published in GMD, which is also available on the GMD website in the 'Manuscript Types' section:

http://www.geoscientific-model-development.net/submission/manuscript_types.html

[Figure]

In particular, please note that for your paper, the following requirements have not been met in the Discussions paper:

- "The main paper must give the model name and version number (or other unique identifier) in the title."

- "If the model development relates to a single model then the model name and the version number must be included in the title of the paper. If the main intention of an article is to make a general (i.e. model independent) statement about the usefulness of a new development, but the usefulness is shown with the help of one specific model, the model name and version number must be stated in the title. The title could have a form such as, "Title outlining amazing generic advance: a case study with Model XXX (version Y)"."

- "Code must be published on a persistent public archive with a unique identifier for the exact model version described in the paper or uploaded to the supplement, unless this is impossible for reasons beyond the control of authors. All papers must include a section, at the end of the paper, entitled "Code availability". Here, either instructions for obtaining the code, or the reasons why the code is not available should be clearly stated. It is preferred for the code to be uploaded as a supplement or to be made available at a data repository with an associated DOI (digital object identifier) for the exact model version described in the paper. Alternatively, for established models, there may be an existing means of accessing the code through a particular system. In this case, there must exist a means of permanently accessing the precise model version described in the paper. In some cases, authors may prefer to put models on their own website, or to act as a point of contact for obtaining the code. Given the impermanence of websites and email addresses, this is not encouraged, and authors should consider improving the availability with a more permanent arrangement. Making code available through personal websites or via email contact to the authors is not sufficient.

After the paper is accepted the model archive should be updated to include a link to the GMD paper."

Thus, WRF including the applied model version need to be named in the title of the paper.

Even more important, it is not sufficient to state that all things are available from the author upon request. We are an open access open source journal. Thus whereever not prevented by licenses, the code and data needs to be made publicly available. If this is not possible, you have to state explicitly why.

Yours,

Astrid Kerkweg

---

## Author Comment (AC1) · 27 Aug 2020

According to the requests by Astrid Kerkweg, Executive editor of GMD, we added the model name and version number in the title of the main paper and of the accompanying Supplementary Material. As well, we have made publicly available all data (netcdf files) and relevant codes (scripts for data analysis and WRF namelists) for reproducing our study, being accessible at http://doi.org/10.23728/b2share.a2af68d0b6514cc5a29f084d42bda1c6. In consequence, we changed the 'Code and data availability' section of the paper.

---

## Referee Comment (RC1) · Anonymous Referee #1 · 14 Sep 2020

This paper quantifies a present day and future reduction in summertime solar radiation at the surface due to aerosol and aerosol-cloud interactions over Europe using WRF as a regional climate model (RCM). Previous work has used static aerosol concentrations to quantify insolation reductions due to aerosol, while this study uses online dynamic aerosols through the GOCART module in WRF-Chem.

Overall, this paper is missing interpretations of key physical processes, model validation, may have a flawed model design, and does not fall under the purview of GMD. As the paper currently stands, my recommendation is "reject." To enter major-revisions territory, significant changes to the model setup, experimental design, and analysis

would be necessary.

Major Comments

1. I do not believe that with the current model namelist settings there is a realistic representation of aerosol-cloud-interactions (ACI). It is my understanding that WRF-Chem requires aqueous-phase chemistry combined with a modal / sectional aerosol scheme (MOSAIC or MADE/SORGAM) to model ACI. This experiment uses GOCART for aerosol, which is single moment in mass, whereas double moment in mass + number is required for ACI studies. I point the authors to the WRF-Chem User's Guide, which has a section on setting up the model for ACI.

I am familiar with the Thompson & Eidhammer (2014) aerosol-aware microphysics (MP), which backs out aerosol number information from the mass-only GOCART values via a lognormal aerosol distribution assumption. After digging around in the source code, I believe the module_mixactivate.F might do something similar for the Lin-GOCART setup. However, the specifics and whether or not and how the model is doing this transformation (or defaulting to a prescribed constant number when a sectional aerosol model isn't found) needs to be confirmed by the authors. This mass to number conversion does not make a scheme double moment because number is not a prognostic variable: it's inferred. This single moment approach is not enough to study ACI in a dynamical framework.

2. It's not considered ACI by the community to run a non-chemistry WRF simulation with a prescribed constant CCN number (single moment cloud) to a simulation with dynamic aerosol (double moment cloud). The change in moments and the change in CCN are intertwined and you cannot deconvolve these changes from each other. It is more realistic to run two WRF-Chem simulations with scale emissions and to run everything in double-moment.

3. There is a difference in which autoconversion scheme is called between progn=0 and progn=1 in the Lin-MP (single moment vs double moment cloud). Some of the ACI

attributed here is from the difference in the representation of autoconversion and not actually from ACI. This scheme change can be significant – see Liu et al. 2005 in GRL.

4. This manuscript makes no attempt to attribute the results to physical processes for ACI. Why are we seeing these results? What microphysical or environmental processes are actually causing the change in cloudiness? It's not enough to simply state that the change occurs. Most of the results section of the manuscript is describing what is on the plots and not interpreting the physics.

5. The WRF simulations are compared to the coarse GCM for validation. Why not compare them (at least in the present-day scenario) to reanalysis that is run at higher resolution? There is no validation of the model against observations. At least reanalysis incorporates observations and is a start for validation.

6. It is not clear what value is added by including gas-phase chemistry in these simulations. The pathways that contribute to aerosol are not explained.

7. Breaking up the contribution to AOD and to ACI by aerosol type would be useful (e.g. carbon and dust will not have the same effect on CCN number as sulfates).

8. The overarching narrative of the paper is not clear. Is the point to compare RCM static aerosol to RCM dynamic aerosol? To assess the value added from moving from GCM dynamic aerosol to RCM dynamic aerosol? By the end of the paper, I had completely lost track of science question.

9. The English needs reviewing throughout the manuscript. More time is needed to revise the grammar and spellings than can be provided here.

Specific Comments (page),[lines]

1. (2),[34-35] – What are GCMs modeling dynamically that RCMs are not?

2. (4),[84] – Why WRF-3.6.1? It's on version 4.2.1 now. Why such an old version?

3. (4),[87] – Why use GCM boundary conditions and not reanalysis? The CORDEX

protocol suggests running the present-day experiments in the "perfect boundary condition experiment mode" with reanalysis and then running the future RCP scenarios with GCM boundary conditions.

4. (5),[98-99] – What is meant here by aerosol radiation is an external forcing?

5. (5),[130-131] – The manuscript needs to stand on its own. If the focus of the paper is on ACI, then ACI in the model and model limitations in representing ACI within the setup and the resolution need to be described in full detail here.

6. (5),[139-141] – See major comment #1

7. (6),[144-146] – So the data was subset by the researchers for non-cloudy days? Radiation code often outputs clear-sky values. Why not use that to ensure a constant data stream?

8. (6),[144-146] – Is clear-sky only for that grid box where the threshold is met or is more data around those grid boxes removed?

9. (6),[144-146] – Why do the clear sky values matter? Need to tell the readers why these are useful metrics to include.

10. (6),[154] – I'm lost in how averaging was done throughout this section and which time scales we are looking at. Are these a daily daytime mean that was then averaged into summertime means? Was the data filtered to exclude nighttime values?

11. (6),[154] - The methodology for calculating the correlations, (especially temporal correlations) needs to be described.

12. (6),[156-158] – Wouldn't the solar industry also be interested in effects under reduced solar output times (i.e. winter)?

13. (6),[156-158] – The direct radiative effect is strongest in summer, but what about the indirect effect?

14. (7),[172-173] – Why is the spatial pattern in the response occurring? Why do some parts have an increase and some have a decrease? What is happening microphysically? Is it a difference in aerosol type that is causing this?

15. (7),[177-181] – The wording here is confusing. Differences of what exactly? Is the point to say that CTT reduces RSDS more than AOD? This needs more explanation.

16. (7),[184-185] – I don't see how the explanation in the previous paragraph proves this connection.

17. (8),[204-205] – How does the previous point imply orbital issues or water vapor? The link is not clear.

18. (8),[206] – There is no transition into now looking at the future projections. Maybe split up into Section 3.A for present-day and 3.B for future.

19. (8),[219] – Where was this specified in Section 2?

20. (9),[241-243] – 5% compared to what? GCM? No aerosol?

21. (9),[247-248] – Why are RSDS and cloudiness not linked? What are the physics here?

22. (9),[249-250] – What does this statement mean?

23. (9),[250-253] – Why is this conclusion significant in a broader context?

---

## Referee Comment (RC2) · Anonymous Referee #2 · 18 Sep 2020

This manuscript, submitted to Geoscientific Model Development, presents a sensitivity study on the role of dynamic aerosols in regional climate simulations over Europe, carried out with the WRF model. The authors consider both present and future simulations, and discuss the role of aerosol-radiation and aerosol-cloud interactions respectively. They conclude that the response of downwelling surface shortwave radiation (rsds) to aerosols is mainly driven by the impact of aerosols on cloudiness. Overall this question is very interesting and needs to be studied, I found the present manuscript presents major problems of methodology, that is the reason why I would suggest not to publish it in GMD.

[Figure]

Main comments:

- The authors are ambiguous about the objective of their study, to begin with the title. I do not understand if they want (1) to show the added values of representing interactive dynamic aerosols in regional climate simulations, compared to regional climate simulations with climatological aerosols, or (2) if they want to show the mean impact of aerosols in regional climate simulations compared to simulations which would not have any aerosols. Given the title, I was expected the first option, which is a very interesting question, not very much documented in literature, but this requires a rigorous protocol in which we compare regional climate simulations with the same aerosol content on average. This is not the case here. So I suppose the authors were in the second option, which is much less interesting, as it has already been studied in different publications. In that case, I suggest to remove the word dynamic from the title, and avoid overly affirmative expressions such as "a reduction about 5% in RSDS was found when aerosols are dynamically solved by the RCM".

- Another major concern about this study is the fact that the authors draw conclusions on the impact of aerosols on rsds future evolution, while they keep constant anthropogenic emissions in their future simulation. The authors are aware of discrepancies in the rsds future evolution between global and regional climate simulations, which could be due to the use of constant aerosols in RCMs contrary to GCMs (Boé et al. 2020). That is the reason why I do not understand the authors keep anthropogenic aerosol emissions constant in future simulations, while they should evolve as in the GCM simulation.

- The last major concern is about the RCM used in this study. The version of WRF used here, namely 3.6.1 is quite old (reference paper from 2008), and above all a precise description of how aerosols and their effects on climate are represented is missing. For example, I wonder what aerosol climatology is used in the BASE simulation (if it is not zero). I am also very worried about the very low values of summer AOD shown in Figure 1g-h, which shows that WRF clearly underestimates AOD over Europe. WRF

values range from 0.05 to 0.09 over Europe, while observations typically range from 0.1 to 0.2 (Papadimas et al. 2008, Nabat et al. 2013, Schultze and Rockel 2018). That could lead to an underestimation of aerosol effects. In such a study, an evaluation of AOD (even brief) is needed in order to ensure the consistency of the results.

Other comments :

- page 2 line 31: land use change is not specific to regional climate simulations, I think it is even more used in global climate simulations.

- page 3 lines 57-63: please avoid such long lists of references, and clarify the conclusions of each of them

- page 3 lines 70-71: "which still remain largely a mystery". Other studies such as Giorgi et al. (2016), Sørland et al. (2018) and Boé et al. (2020) have also underlined differences between RCMs and GCMs in future projections. The role of aerosols is even discussed in Boé et al. (2020), which should be mentioned here.

- page 5 lines 140-141: it is not clear for me how aerosol-cloud interactions are represented in the simulations.

-page 6 lines 144-146: This way of calculating clear-sky variables in simulations is not common in modeling studies. It would be appropriate for a comparison to observations (it is exactly how satellites do for example), but in models, you generally compute clear-sky variables at each time step, removing clouds in radiative transfer. This would avoid the numerous missing values.

- page 6 section 3: This section should be divided in several sub-sections, with more precise titles than only "Results".

- page 6 line 165: "The inclusion of interactive aerosols reduce the JJA mean values of RSDS". This is typically an example of my first main comment. This decrease in rsds is likely due to the mean effect of aerosols, and not their interactive pattern.

- page 7 line 172: "ARI and ACI lead to more cloudiness in central and northern regions". This is not really the case when looking at the figure.

- page 7 lines 184-187: This conclusion is not justified.

- Figures 1-4: From my point of view it would be easier to understand to have differences in absolute values rather than in percentages. Indeed, I suspect here we look at very low values which could be unsignificant.

- Figures 1-4: Why consider only land points ? It would be interesting to show also ocean points on figures.

- Figure 3: When comparing the evolution of rsds, cct and aod in the simulations, I suspect a possible bug in the figure or in the simulation. Indeed, the strong decrease in rsds in northern latitudes (for example in Iceland), is neither explained by cct nor by aod.

- Page 8 lines 218-219: If "the anthropogenic component is disregarded", there should be no possible conclusion on the future evolution of rsds.

- The manuscript suffers from many typographical and English spelling errors that need to be corrected.

---

## Referee Comment (RC3) · Anonymous Referee #3 · 28 Sep 2020

This manuscript, submitted to Geoscientific Model Development, aims to identify the role of interactively modeling aerosol in regional climate simulations over Europe, by conducting a sensitivity study with the WRF model. The focus is on solar radiation at the surface during summer. Both a present and a future period are considered. Changes in cloudiness are presented as the main driver of the changes in solar radiation. There are some interesting features in this study, such as long simulations with the WRF-Chem model using interactive aerosol that are computationally demanding. However, I believe that the main problem is that the aim of the study is not actually addressed. I believe that separating the "interactive" part of aerosol modeling and making general comments about it is not possible in the current study. Thus it is a problem of

methodology and structuring of the whole manuscript. Moreover I believe that a significant clarification is need in the current methodology regarding the BASE simulation that is the basis for comparison. I would hesitate to recommend it for publication in its current form. However, I believe that it could stand as a sensitivity study aiming to describe the impact of the specific model and aerosol treatments used. I would suggest major revisions regarding: the aims of the study, including a validation, possibly changing the analysis under clear-sky conditions, clarifying the aerosol treatment in the simulations. In the end, I think the study could provide some interesting points to the community.

Major comments:

1. One of my major concerns is that the nature of the BASE experiment is not clear to me. It is stated that it works with a specific aerosol concentration and that "the aerosol radiative effect is assumed to come as an external forcing." I am not sure what this means. Does the BASE experiment let these aerosols interact with radiation? In this case the AOD field needs to be shown. Or their only impact is that they are just used by the microphysics to facilitate cloud formation? In any case, the nature of aerosol in the BASE experiment needs to be clearly stated so that the reader understands the results of the comparison. Moreover if BASE has an AOD that interacts with radiation, how much does it differ from the AOD of ARI and ACI? Are the differences between BASE and these simulations attributed to the difference in AOD and not to the introduction of dynamic aerosol?

2. It is very interesting to try and identify the impact of interactively modeled aerosols. However, I am not sure that this is achieved in the study. You can make a statement that, for example, the ARI experiment that uses "this specific" interactive aerosol treatment in WRF-Chem has "this specific impact" on radiation. This statement could be useful to the community as a sensitivity study of the model and aerosol scheme. However, I do not think that you can attribute this impact only to the "interactive" part. Probably, a first step towards that direction would be to have additional experiments

enabling aerosol-radiation and cloud interactions using static aerosol fields with the same mean AOD as the ones in ARI and ACI.

3. I believe a validation (even a quick one) of the simulations, especially regarding rsds and AOD, should be part of the study in order to assert that they do capture the basic patterns of the examined variables. I do understand that they are compared against the GCM (and that the GCM has been probably validated), but still a validation would make the results more robust.

4. The methodology to calculate Clear sky conditions was a bit unusual to me. I am aware that the radiation code in WRF (and I think this is the case for version 3.6.1) provides the clear-sky radiation at every time step simultaneously with rsds. It would probably be better to use that feature. I also have a question regarding the methodology. It is stated (page 6, 150-152) that in order to consider a specific grid point in the analysis you need to have at least 15 records per period that are not missing values. Ok so far. It is stated (page 6, lines 153-154) that "(which, according to our methodology, would occur only if all days within a summer season have CTT values >1%)." So, if I understand correctly even if one day within a summer season has a CCT value <1, that summer season gains a valid value based only on that day and is considered in the analysis?

5. The use of no time evolving anthropogenic aerosol in the future period by ARI and ACI experiments is not ideal. It is good that this deficiency is stated in the manuscript (page 8, line 218). Moreover, it would be interesting to see what are the rsds differences between the GCM and ARI/ACI for the future period.

Minor comments:

-Page 1, line 20 "reduction about 5% in RSDS was found when aerosols are dynamically solved". This is compared to BASE? It must be clearly stated.

-Page 2, line 33 The phrase "all about cumulus" I believe should be clarified a bit better.

[Figure]

Is this about convective phenomena, the cloud fraction scheme or both?

-Page 4 lines 97-98. In the BASE experiment "the by-default WRF setup was used, which considers 250 cloud condensation nuclei per cm3 to form clouds". I think the term "by-defalut" might be a bit misleading. I understand that this concentration of CCN is probably related to the Lin microphysics scheme used in the experiments and this should be stated.

-I do not understand how ACI (page 5, lines139-141) works. What is meant by "Although this WRF-Chem version (3.6.1) does not allow a full coupling with aerosol-cloud interactions..."? I believe it should be clearly stated which are the parts of the aerosol-clouds interactions that are missing. Also I think it should be stated to which variables the single and double moment treatment is applied.

-I believe it is useful to know which statistical test is used (t-set, non parametric Mann-Whitney...) to determine statistical significance.

-Total cloud cover values over southern Europe in summer are usually small. Thus, the changes in CCT between the experiments could be in some cases negligible but the relative (percentage) change could be inflated. I believe this should be stated in the manuscript. Also, it would be interesting to see a plot with the plain difference in CCT between experiments in the supplement.

-Page 7, lines 185-186. "Contrary, the effect of interactive aerosols schemes..." The way it is written gives the impression that the authors are talking about interactive schemes in general. I think it would be better to avoid generalizing the results of this specific sensitivity study.

-Page 8, lines 209-210. "These latter are more widespread in ARI than in BASE, which makes the ARI pattern the most similar to the change pattern from the GCM". I do not clearly see this in Figure3.

Technical corrections:

Page 7 line 183 "varables" -> variables

Page 7, line 188 I am not aware of the word "devanishes". Could this be a spelling mistake?

Page 10, line 274 experimts -> experiments

Page 1, line25 much more softer -> much softer
* * *

---

## Author Comment (AC2) · 15 Oct 2020

Authors' response

**This paper quantifies a present day and future reduction in summertime solar radiation at the surface due to aerosol and aerosol-cloud interactions over Europe using WRF as a regional climate model (RCM). Previous work has used static aerosol concentrations to quantify insolation reductions due to aerosol, while this study uses online dynamic aerosols through the GOCART module in WRF-Chem.**

**Overall, this paper is missing interpretations of key physical processes, model validation, may have a flawed model design, and does not fall under the purview of GMD. As the paper currently stands, my recommendation is "reject." To enter major-revisions territory, significant changes to the model setup, experimental design, and analysis would be necessary.**

We do thank the reviewer for the time devoted to read and thoughtfully comment on our work. Below we provide detailed answers to each comment, hoping to have been clear enough in our explanations. Attending these comments and the ones posted by the other reviewers and the Editor, the new version of the manuscript:

1 – Has been entirely revised by a native speaker in order to improve the redaction.

2 – Has a new title. The change intends to avoid that the reader interprets that we are comparing simulations with dynamic vs. static aerosols. The new title is:

"Sensitivity of surface solar radiation to aerosol-radiation and aerosol-cloud interactions over Europe in WRFv3.6.1 climatic runs with fully interactive aerosols"

In this line, we have made an effort to make the scientific purpose of the manuscript clearer throughout the whole text.

3 – Includes further details and arguments on the experimental set-up and the methodology. Section 2 has been divided into 3 subsections.

4 – The formerly labeled as ACI simulations are now named ARCI to emphasize that these include both aerosol-radiation and aerosol-clouds interactions.

5 – Includes a brief validation exercise. We now face the outputs of our simulations with ERA5.

6 – Includes two subsections within the Results section: one for the historical simulations and another for the future projections.

7 – Includes a deeper discussion of the results attending several comments by the reviewers, e.g.:

- the activation of the autoconversion scheme in the ARCI simulations hampers a direct attribution of the signals to the aerosol-cloud interactions from a physical point of view (the attribution can be made, from a modeling point of view, to the activation of these interactions in the model);

- the fact that we kept constant the anthropogenic aerosol emissions in future simulations permits to better isolate the signals from the aerosol-radiation-cloud interactions due to the so-called climate change penalty alone, while reduces the reliability of the future projections obtained;

- the signals obtained for different seasons (additional analysis is provided as Supplementary Material);

8 – Includes a link where all the data and codes to reproduce our study have been made publicly available: http://doi.org/10.23728/b2share.682b1c6311134b36a18f59a99a443afd.

We are confident that these major changes have improved significantly the manuscript and provides a larger support to its key findings.

We must also notice that we used wrong AOD values in the previous version of the paper, as it was noted by the reviewer2. These had been computed from the TAUAER3 and TAUAER4 variables, which do exhibit a weird evolution along the year. After inspection, we figured out that these and the EXTCOF55 variables had been wrongly recorded in the wrfout files (not new, apparently, see e.g.: https://forum.mmm.ucar.edu/phpBB3/viewtopic.php?t=9313&p=17464). So we have now adopted an alternative method to compute AOD following Palacios-Peña et al (2020), where, in fact, the representation of AOD by these model configurations (ARI and ARCI) were deeply evaluated. The new AOD files were estimated using the reconstructed mass-extinction method (Malm et al 1994) from the well-recorded concentrations of the various aerosol species in the wrfout files, namely: black carbon, organic carbon, dust and sea salt. Sulfates were estimated from SO2 and OH recorded concentrations using the same kinetic reaction as the one implemented in the RACM-KPP module. We want to remark that the mistake occurered during the postprocessing of the wrfout files, while WRF-Chem run satisfactorily. These wrfout files were removed after postprocessed, so we have now generated a sample one (using the ARI configuration) and uploaded it for checking together with all the other data files. Importantly, this change in the methodology for estimating AOD values did not alter the the overall results of the paper.

Regarding the interest of our work for the GMD audience, it should be noted that we submit it to the inter-journal Special Issue *Chemistry–Climate Modelling Initiative*. Although the managing Editor should have agreed it is within the scope of the journal and the Special Issue, we would be open to move it from GMD to a counterpart journal.

**Major Comments**

**1. I do not believe that with the current model namelist settings there is a realistic representation of aerosol-cloud-interactions (ACI). It is my understanding that WRF-Chem requires aqueous-phase chemistry combined with a modal/sectional aerosol scheme (MOSAIC or MADE/SORGAM) to model ACI. This experiment uses GOCART for aerosol, which is single moment in mass, whereas double moment in mass + number is required for ACI studies. I point the authors to the WRF-Chem User's Guide, which has a section on setting up the model for ACI.**

**I am familiar with the Thompson & Eidhammer (2014) aerosol-aware microphysics (MP), which backs out aerosol number information from the mass-only GOCART values via a lognormal aerosol distribution assumption. After digging around in the source code, I believe the module_mixactivate.F might do something similar for the Lin-GOCART setup. However, the specifics and whether or not and how the model is doing this transformation (or defaulting to a prescribed constant number when a sectional aerosol model isn't found) needs to be confirmed by the authors. This mass to number conversion does not make a scheme double moment because number is not a prognostic variable: it's inferred. This single moment approach is not enough to study ACI in a dynamical framework.**

This is a well-argued concern. However, we confirm that although the microphysics implemented in the simulations rely on the Lin scheme, this single moment scheme turns into a double moment scheme in the simulations denoted as ACI. See details on how ARCI are implemented in the simulations in the response to the reviewer's comment #2 below. We have also added these details in the manuscript.

**2. It's not considered ACI by the community to run a non-chemistry WRF simulation with a prescribed constant CCN number (single moment cloud) to a simulation with dynamic aerosol (double moment cloud). The change in moments and the change in CCN are intertwined and you cannot deconvolve these changes from each other. It is more realistic to run two WRF-Chem simulations with scale emissions and to run everything in double-moment.**

As aforementioned, the Lin scheme is a single moment scheme based on Lin et al. (1983), including some modifications, such as saturation adjustment (Tao et al. 1989) and ice sedimentation, which is related to the sedimentation of small ice crystals (Mitchell et al. 2008). It includes six classes of hydrometeors: water vapour, cloud water, rain, cloud ice, snow, and graupel. This scheme was one of the first to parameterize snow, graupel, and mixed-phase processes (such as the Bergeron process and hail growth by riming), and it has been widely used in numerical weather studies.

The one-moment microphysical scheme is, effectively, unsuitable for assessing the aerosol-clouds interactions as it only predicts the mass of cloud droplets and does not represent the number or concentration of cloud droplets (Li et al. 2008). The prediction of two moments provides a more robust treatment of the particle size distributions, which is key for computing the microphysical process rates and cloud/precipitation evolution. Therefore, prediction of additional moments allows greater flexibility in representing size distributions and hence microphysical process rates.

In this sense, although the Lin microphysics is presented as a single moment scheme, the WRF-Chem model allows to transform the single into a double moment scheme. A prognostic treatment of cloud droplet number was added (Ghan et al. 1997), which treats water vapour and cloud water, rain, cloud ice, snow, and graupel. The autoconversion of cloud droplets to rain droplets depends on droplet number (Liu et al. 2005). Droplet-number nucleation and (complete) evaporation rates correspond to the aerosol activation and resuspension rates. Ice nuclei based on predicted particulates are not treated. However, ice clouds are included via the prescribed ice nuclei distribution following the Lin scheme. Finally, the interactions of clouds and incoming solar radiation have been implemented by linking simulated cloud droplet number with the Goddard shortwave radiation scheme, representing the first indirect effect, and with Lin microphysics, which represents the second indirect effect (Skamarock et al. 2008). Therefore, droplet number will affect both the calculated droplet mean radius and cloud optical depth.

References:

Ghan, S. J., Leung, L. R., Easter, R. C., & Abdul Razzak, H. (1997). Prediction of cloud droplet number in a general circulation model. *Journal of Geophysical Research: Atmospheres, 102*(D18), 21777-21794.

Li, G., Wang, Y., & Zhang, R. (2008). Implementation of a two moment bulk microphysics scheme to the WRF model to investigate aerosol cloud interaction. *Journal of Geophysical Research: Atmospheres, 113*(D15).

Lin, Y. L., Farley, R. D., & Orville, H. D. (1983). Bulk parameterization of the snow field in a cloud model. Journal of Climate and Applied Meteorology, 22(6), 1065-1092.

Liu, Y., Daum, P. H., & McGraw, R. L. (2005). Size truncation effect, threshold behavior, and a new type of autoconversion parameterization. *Geophysical research letters*, *32*(11).

Mitchell, D. L., Rasch, P., Ivanova, D., McFarquhar, G., & Nousiainen, T. (2008). Impact of small ice crystal assumptions on ice sedimentation rates in cirrus clouds and GCM simulations. *Geophysical research letters*, *35*(9).

Skamarock, W.C.; Klemp, J.B.; Dudhia, J.; Gill, D.O.; Barker, D.M.; Wang, W.; Powers, J.G. A Description of the Advanced Research WRF Version 3; Technical Report NCAR Tech. Note TN-475+STR; NCAR: Boulder, CO, USA, 2008.

Tao, W. K., Simpson, J., & McCumber, M. (1989). An ice-water saturation adjustment. *Monthly Weather Review*, *117*(1), 231-235.

**3. There is a difference in which autoconversion scheme is called between progn=0 and progn=1 in the Lin-MP (single moment vs double moment cloud). Some of the ACI attributed here is from the difference in the representation of autoconversion and not actually from ACI. This scheme change can be significant – see Liu et al. 2005 in GRL.**

The reviewer is totally right. The autoconversion scheme activated with progn=1 (ARCI simulations), so that cloud droplets can turn into rain droplets, is different to the autoconversion scheme called with progn=0 (ARI simulations). Henceforth, this change in the flags of WRF-Chem configuration can lead to ARCI-ARI differences that cannot necessarily be attributed to aerosol-cloud interactions from a physical point of view, but also to different processes and schemes that play a role when progn flag is changed from 0 to 1. In this same sense, the activation of the aerosol-cloud interactions requires further changes in the model configuration (as compared to the configuration used for the simulations labeled as ARI) beyond the autoconversion scheme (e.g. activation of aqueous chemistry or wet scavenging processes), that could also have an added impact to the effect that can be strictly attributed to the aerosol-cloud interactions. However, the encoding of WRF-Chem model imposes that ARI experiments should be performed with progn=0 in order not to allow an on-line calculation of cloud condensation nuclei, while ARCI experiments should be run with progn=1 if the on-line estimations of aerosols wants to be used not only for the radiative balance, but also for CCN (which change between progn=0 and progn=1 simulations). This is true not only with the Lin scheme used here, but also with the Morrison microphysics parametrization (the other scheme available including a double-moment mycrophysics). Therefore, ARCI-ARI differences can not be strictly attributed to the aerosol-cloud interactions from a purely physical point of view, but to the activation of the aerosol-cloud interactions from a modeling point of view (that involves several modifications, including the autoconversion process as stated by the reviewer). All these unavoidable changes are intrinsic to the definition of the flags leading to the representation of aerosol-cloud interactions in WRF-Chem executions. We have now emphasized in the manuscript this aspect of the model configuration and its potential repercussion when interpreting the signals.

**4. This manuscript makes no attempt to attribute the results to physical processes for ACI. Why are we seeing these results? What microphysical or environmental processes are actually causing the change in cloudiness? It's not enough to simply state that the change occurs. Most of the results section of the manuscript is describing what is on the plots and not interpreting the physics.**

We now make more emphasis in attributing the signals to the direct, semi-direct and indirect aerosols effects.

**5. The WRF simulations are compared to the coarse GCM for validation. Why not compare them (at least in the present-day scenario) to reanalysis that is run at higher resolution? There is no validation of the model against observations. At least reanalysis incorporates observations and is a start for validation.**

The manuscript now includes a brief comparison of the present-day simulations with ERA5.

**6. It is not clear what value is added by including gas-phase chemistry in these simulations. The pathways that contribute to aerosol are not explained.**

Chemical reactions in the GOCART model include several oxidation processes by the three main oxidants in the troposphere: OH, NO3, and O3. The OH radical dominates oxidation during the daytime, but at night its concentration drops and NO3 becomes the primary oxidant (Archer-Nicholls et al., 2014). So, the oxidation pathways represented in GOCART include: (a) the dimethyl sulphide (DMS) oxidation by the hydroxyl radical (OH) during the day to form sulphur dioxide ($SO_2$) and methanesulfonic acid (MSA); (b) the oxidation by nitrate radicals (NO3) at night to form $SO_2$; and (c) the $SO_2$ oxidation by OH in air and by $H_2O_2$ and tropospheric ozone ($O_3$) in clouds (aqueous chemistry) to form sulphate (Chin et al., 2000). Henceforth, the skilful characterization of gas-phase radicals such as OH and NO3 or compounds like $O_3$ is essential for the representation of oxidation pathways in the atmosphere leading to the formation of secondary aerosols (Jiménez et al., 2003). Therefore, in this contribution the RACM (Stockwell et al., 1997; Geiger et al., 2003) mechanism has been coupled to GOCART trough the kinetics pre-processor (KPP) in WRF-Chem in order to provide the concentrations of radical and gas-phase pollutants needed by the GOCART aerosol model. We have added this explanation in the text.

References:

Archer-Nicholls, S., Lowe, D., Utembe, S., Allan, J., Zaveri, R. A., Fast, J. D., Hodnebrog, Ø., Denier van der Gon, H., McFiggans, G. (2014). Gaseous chemistry and aerosol mechanism developments for version 3.5.1 of the online regional model, WRF-Chem. *Geoscientific Model Development*, 7, 2557–2579.

Chin, M., Rood, R.B., Lin. S.-J., Müller, J.-F., Thompson, M. (2000). Atmospheric sulfur cycle simulated in the global model GOCART: Model description and global properties. *Journal of Geophysical Research*, 105(D20), 24671-24687.

Geiger, H., Barnes, I., Bejan, I., Benter, T., & Spittler, M. (2003). The tropospheric degradation of isoprene: an updated module for the regional atmospheric chemistry mechanism. *Atmospheric Environment*, 37, 1503 - 1519.

Jiménez, P., Baldasano, J.M., Dabdub, D. (2003). Comparison of photochemical mechanisms for air quality modeling. *Atmospheric Environment*, 37, 4179-4194.

Stockwell, W. R., Kirchner, F., Kuhn, M., & Seefeld, S. (1997). A new mechanism for regional atmospheric chemistry modeling. *Journal of Geophysical Research: Atmospheres*, 102, 25 847 - 25 879.

**7. Breaking up the contribution to AOD and to ACI by aerosol type would be useful (e.g. carbon and dust will not have the same effect on CCN number as sulfates).**

We can not afford to disentangle the contribution of each aerosol type to the effects attributed to the activation of the aerosol-clouds interactions in the simulations. It would require to run the ARCI simulations including only one of the aerosol species (5 in GOCART) at once. But each ARCI run takes months to be performed due to its expensive computational cost. So this is not a feasible task for us in a reasonably time.

On the other hand, we found that the main driver for the differences between the runs with aerosols and the runs without them is cloudiness, while the AOD plays a secondary role, which justifies the low attention paid to disentangling the contribution of each aerosol species to the AOD.

**8. The overarching narrative of the paper is not clear. Is the point to compare RCM static aerosol to RCM dynamic aerosol? To assess the value added from moving from GCM dynamic aerosol to RCM dynamic aerosol? By the end of the paper, I had completely lost track of science question.**

We have made an effort to make it clearer, starting by the title. The point is to evince the impact of aerosol-radiation and aerosol-cloud interactions in WRF runs performed with dynamic aerosols by comparison with baseline WRF runs performed without aerosols (nor dynamic, nor static). The baseline set-up is the most common one in the currently available portfolio of regional climate change scenarios provided under the umbrella of benchmark initiatives such as Euro-Cordex.

**9. The English needs reviewing throughout the manuscript. More time is needed to revise the grammar and spellings than can be provided here.**

Done with the help of a native speaker.

**Specific Comments (page),[lines]**

**1. (2),[34-35] – What are GCMs modeling dynamically that RCMs are not?**

It was said: "This is the case of the atmospheric aerosols concentration and their multiple non-linear interactions (eg. Taylor et al 2012 vs. Ruti et al 2016), the so-called aerosol-radiation and aerosol-cloud interactions (Boucher 2015)."

**2. (4),[84] – Why WRF-3.6.1? It's on version 4.2.1 now. Why such an old version?**

Some of the simulations included in this work were performed time ago. Others have been performed more recently, but we decided to use the same WRF version to be sure of being comparing the same "thing". At that time, when first simulations were carried out, the last stable version of WRF was the 3.6.1. In any case, the physics of the model is the same, no matter of its version.

**3. (4),[87] – Why use GCM boundary conditions and not reanalysis? The CORDEX protocol suggests running the present-day experiments in the "perfect boundary condition experiment mode" with reanalysis and then running the future RCP scenarios with GCM boundary conditions.**

We used GCM boundary conditions because we were to also asses impacts on future projections (not only sensitivity under present climate). Nonetheless, we have at our disposal a set of identical runs (BASE, ARI and ARCI configurations) using the reanalysis ERA20C and initial and boundary conditions. We are aware that the Euro-Cordex protocol establishes the use of Era-Interim as "perfect boundary conditions", but we needed a longer period (for reasons that are irrelevant here)

and used ERA20C instead. The results from these simulations are attached below. These basically resemble those already included in the paper. Therefore, we decided not to include them in the paper for the sake of brevity.

**4. (5),[98-99] – What is meant here by aerosol radiation is an external forcing?**

This sentence was misleading and has been removed.

**5. (5),[130-131] – The manuscript needs to stand on its own. If the focus of the paper is on ACI, then ACI in the model and model limitations in representing ACI within the setup and the resolution need to be described in full detail here.**

We have extended the description of the model configuration and discussed about it, as explained above.

**6. (5),[139-141] – See major comment #1**

We now explain better this aspect of the model configuration.

**7. (6),[144-146] – So the data was subset by the researchers for non-cloudy days? Radiation code often outputs clear-sky values. Why not use that to ensure a constant data stream?**

Unfortunately, we did not save clear-sky values from the model outputs, so we needed to adopt an alternative methodology.

**8. (6),[144-146] – Is clear-sky only for that grid box where the threshold is met or is more data around those grid boxes removed?**

The criteria is applied at the grid-box level, for each grid-box individually and independently. This has been also clarified in the text.

**9. (6),[144-146] – Why do the clear sky values matter? Need to tell the readers why these are useful metrics to include.**

We have included: "The analysis in absence of cloudiness will tell us about the relevance of the direct radiative effect of aerosols."

**10. (6),[154] – I'm lost in how averaging was done throughout this section and which time scales we are looking at. Are these a daily daytime mean that was then averaged into summertime means? Was the data filtered to exclude nighttime values?**

We have better explain it in section 2 (in the new subsection 2.3 – Data and methods). We simply averaged over all the JJA (or either season) records in the series.

**11. (6),[154] - The methodology for calculating the correlations, (especially temporal correlations) needs to be described.**

Done in section 2 (in the new subsection 2.3 – Data and methods). The codes used are also made available.

**12. (6),[156-158] – Wouldn't the solar industry also be interested in effects under reduced solar output times (i.e. winter)?**

We now included winter plots in Supp. Material and discussed about the seasonal dependence of our results.

**13. (6),[156-158] – The direct radiative effect is strongest in summer, but what about the indirect effect?**

Our results do support the key role of the indirect aerosols effect in summer indeed.

**14. (7),[172-173] – Why is the spatial pattern in the response occurring? Why do some parts have an increase and some have a decrease? What is happening microphysically? Is it a difference in aerosol type that is causing this?**

The increase in cloudiness in central and northern regions in ARI and ARCI simulations as compared to BASE could be explained through the following feedback mechanism: the cooling effect of the scattering of radiation by the high presence of sea salt, dust and sulphates over these areas cools down surface temperatures, thus increasing relative humidity and favouring the formation of clouds, which leads to less radiation reaching the surface, thus lower surface temperatures, and so on. Nonetheless, these signals would simply indicate that such a enhancing mechanism prevails over others, such as the semi-direct effect that acts to suppress cloudiness.

The reduction in cloudiness southward also appears in both ARI and ARCI simulations, but it is more evident in ARCI. Therefore, both semi-direct and indirect aerosol effects would tend to diminish cloudiness southward, with the latter (indirect effect) holding the strongest impact. This could be due to the fact that a high presence of large aerosols over southern Europe, both in form of dust or sulphate in our case, hampers the formation of clouds and enhances precipitation (shorter-lived clouds) as long as aerosol-cloud interactions are resolved by the model, which is a plausible explanation especially in the warm season over warm areas (e.g. Lee et al., 2008).

We have added and supported these arguments in the main manuscript.

References:

Lee, S. S., Donner, L. J., Phillips, V. T., & Ming, Y. (2008). Examination of aerosol effects on precipitation in deep convective clouds during the 1997 ARM summer experiment. *Quarterly Journal of the Royal Meteorological Society: A journal of the atmospheric sciences, applied meteorology and physical oceanography, 134*(634), 1201-1220.

**15. (7),[177-181] – The wording here is confusing. Differences of what exactly? Is the point to say that CTT reduces RSDS more than AOD? This needs more explanation.**

Yes, that is the point. We have further developed this part to make it clearer.

**16. (7),[184-185] – I don't see how the explanation in the previous paragraph proves this connection.**

It is supported by the fact that differences between pairs of experiments in CCT correlates more than differences between pairs of experiments in AOD with the differences between pairs of experiments in RSDS. This showed up both, spatially (see s_corr values in Fig 1d-i) and temporally (Supp Fig 5a-f).

**17. (8),[204-205] – How does the previous point imply orbital issues or water vapor? The link is not clear.**

Different days (dates) may have different daytime lengths and different atmospheric compositions (thus different atmospheric optical depth or atmospheric transmissivity) that may mask the AOD effect under clear-sky conditions. We have better explained in the text what we meant.

**18. (8),[206] – There is no transition into now looking at the future projections. Maybe split up into Section 3.A for present-day and 3.B for future.**

Done.

**19. (8),[219] – Where was this specified in Section 2?**

It was said:

"Anthropogenic emissions coming from the Atmospheric Chemistry and Climate Model Intercomparison Project (ACCMIP; Lamarque et al 2010) were kept unchanged in the simulation periods (we considered the 2010 monthly values)."

We have now further emphasized and discussed this feature.

**20. (9),[241-243] – 5% compared to what? GCM? No aerosol?**

Compared to the BASE experiment (without aerosols). Now specified.

**21. (9),[247-248] – Why are RSDS and cloudiness not linked? What are the physics here?**

It was said the opposite: "Differences in RSDS between experiments are in overall good agreement with the differences found in cloudiness"

**22. (9),[249-250] – What does this statement mean?**

That statement was removed. It was certainly confused.

**23. (9),[250-253] – Why is this conclusion significant in a broader context?**

We now argument about the importance of the signals under clear-sky conditions.

**RSDS JJA climatologies for 1991-2010**

[Figure]

RSDS summer climatologies in the present period from ERA20C (a), ERA5 (b) and the ERA20C-driven WRF simulations (d to f); units: Wm$^{-2}$, same colorbar in all cases (the upper one). Panel c depicts relative differences between ERA20C and ERA5, panels g to i between each WRF simulation and ERA20C, and panels j to l between each WRF simulation and ERA5, squared if statistically significant (p<0.05); units: %, same colorbar in all cases (the bottom one).

**RSDS, CCT & AOD JJA climatologies for 1991-2010: differences between experiments**

[Figure]

**(a)** RSDS ARI-BASE

**(b)** RSDS ARCI-BASE

**(c)** RSDS ARCI-ARI

**(d)** CCT ARI-BASE
*s_corr* with (a) = -0.65

**(e)** CCT ARCI-BASE
*s_corr* with (b) = -0.58

**(f)** CCT ARCI-ARI
*s_corr* with (c) = -0.76

**(g)** AOD ARI
*s_corr* with (a) = -0.44

**(h)** AOD ARCI
*s_corr* with (b) = -0.46

**(i)** AOD ARCI-ARI
*s_corr* with (c) = 0.20

Relative differences between the ERA20C-driven WRF simulations in the RSDS (a to c), CCT (d to f) and AOD at 550 nm (g to i) summer (JJA) climatologies in the present period (1991-2010), squared if statistically significant (p<0.05); units: %. Note that panels g and h are referred to the horizontal colorbar just below them and simply represent the AOD summer climatologies in ARI and ARCI respectively. Spatial correlations (*s_corr*) between the patterns in the second and third rows and the respective patterns in the first row are indicated in the headers.

**RSDS$_{cs}$ & AOD$_{cs}$ JJA climatologies for 1991-2010: differences between experiments**

[Figure]

**(a)** RSDS$_{cs}$ ARI-BASE  **(b)** RSDS$_{cs}$ ACI-BASE  **(c)** RSDS$_{cs}$ ACI-ARI  %

**(d)** AOD$_{cs}$ ARI  **(e)** AOD$_{cs}$ ACI  **(f)** AOD$_{cs}$ ACI-ARI
*s_corr* with (a) = -0.24  *s_corr* with (b) = -0.24  *s_corr* with (c) = -0.23  %

Relative differences between the ERA20C-driven WRF simulations in the RSDS$_{cs}$ (a to c) and AOD$_{cs}$ at 550 nm (d to f) summer (JJA) climatologies, this is under clear-sky conditions, in the present period (1991-2010), squared if statistically significant (p<0.05); units: %. Note that panels d and e are referred to the horizontal colorbar just below them and simply represent the AOD summer climatologies in ARI and ARCI respectively. Gray shaded areas depict grid point where less than 75% of the summer mean values in the time series of RSDS$_{cs}$ and AOD$_{cs}$ were not missing values. Spatial correlations (*s_corr*) between the patterns in the second row and the respective patterns in the first row are indicated in the headers.

**Temporal correlations between difference series of RSDS, CCT & AOD: 1991-2010 JJA-mean series**

[Figure]

As obtained from ERA20C-driven WRF experiments, temporal correlations between JJA-mean temporal series of differences in RSDS and CTT (a to c), RSDS and AOD (d to f), and $RSDS_{cs}$ and $AOD_{cs}$ (g to i; gray-shaded areas where the number of time steps in the clear-sky series is below 75% of total time steps) between ARI and BASE (first column), ARCI and BASE (second column), and ARCI and ARI experiments (third column) in the present period (1991-2010). Little stars indicate statistical significance (p<0.05).

---

## Author Comment (AC3) · 15 Oct 2020

Authors' response

**This manuscript, submitted to Geoscientific Model Development, presents a sensitivity study on the role of dynamic aerosols in regional climate simulations over Europe, carried out with the WRF model. The authors consider both present and future simulations, and discuss the role of aerosol-radiation and aerosol-cloud interactions respectively. They conclude that the response of downwelling surface shortwave radiation (rsds) to aerosols is mainly driven by the impact of aerosols on cloudiness. Overall this question is very interesting and needs to be studied, I found the present manuscript presents major problems of methodology, that is the reason why I would suggest not to publish it in GMD.**

We do thank the reviewer for the time devoted to read and thoughtfully comment on our work. Below we provide detailed answers to each comment, hoping to have been clear enough in our explanations. Attending these comments and the ones posted by the other reviewers and the Editor, the new version of the manuscript:

1 – Has been entirely revised by a native speaker in order to improve the redaction.

2 – Has a new title. The change intends to avoid that the reader interprets that we are comparing simulations with dynamic vs. static aerosols. The new title is:

"Sensitivity of surface solar radiation to aerosol-radiation and aerosol-cloud interactions over Europe in WRFv3.6.1 climatic runs with fully interactive aerosols"

In this line, we have made an effort to make the scientific purpose of the manuscript clearer throughout the whole text.

3 – Includes further details and arguments on the experimental set-up and the methodology. Section 2 has been divided into 3 subsections.

4 – The formerly labeled as ACI simulations are now named ARCI to emphasize that these include both aerosol-radiation and aerosol-clouds interactions.

5 – Includes a brief validation exercise. We now face the outputs of our simulations with ERA5.

6 – Includes two subsections within the Results section: one for the historical simulations and another for the future projections.

7 – Includes a deeper discussion of the results attending several comments by the reviewers, e.g.:

- the activation of the autoconversion scheme in the ACI simulations hampers a direct attribution of the signals to the aerosol-cloud interactions from a physical point of view (the attribution can be made, from a modeling point of view, to the activation of these interactions in the model);

- the fact that we kept constant the anthropogenic aerosol emissions in future simulations permits to better isolate the signals from the aerosol-radiation-cloud interactions due to the so-called climate change penalty alone, while reduces the reliability of the future projections obtained;

- the signals obtained for different seasons (additional analysis is provided as Supplementary Material);

8 – Includes a link where all the data and codes to reproduce our study have been made publicly available: http://doi.org/10.23728/b2share.682b1c6311134b36a18f59a99a443afd.

We are confident that these major changes have improved significantly the manuscript and provides a larger support to its key findings.

**Main comments:**

**- The authors are ambiguous about the objective of their study, to begin with the title. I do not understand if they want (1) to show the added values of representing interactive dynamic aerosols in regional climate simulations, compared to regional climate simulations with climatological aerosols, or (2) if they want to show the mean impact of aerosols in regional climate simulations compared to simulations which would not have any aerosols. Given the title, I was expected the first option, which is a very interesting question, not very much documented in literature, but this requires a rigorous protocol in which we compare regional climate simulations with the same aerosol content on average. This is not the case here. So I suppose the authors were in the second option, which is much less interesting, as it has already been studied in different publications. In that case, I suggest to remove the word dynamic from the title, and avoid overly affirmative expressions such as "a reduction about 5% in RSDS was found when aerosols are dynamically solved by the RCM".**

We understand the title may lead to misinterpretations, so we changed it.

As we are comparing RCM outputs from simulations with and without aerosols, we were, effectively, in the second option mentioned by the reviewer. Although there exist previous works in this second option, none of them attempted to unveil the impact of aerosols from a purely modeling approach  such as the one used here, where no prescribed aerosol concentrations are used. So the word 'dynamic' actually makes the difference with previous studies, that's why we included it in the title.

**- Another major concern about this study is the fact that the authors draw conclusions on the impact of aerosols on rsds future evolution, while they keep constant anthropogenic emissions in their future simulation. The authors are aware of discrepancies in the rsds future evolution between global and regional climate simulations, which could be due to the use of constant aerosols in RCMs contrary to GCMs (Boé et al. 2020). That is the reason why I do not understand the authors keep anthropogenic aerosol emissions constant in future simulations, while they should evolve as in the GCM simulation.**

As stated above, this approach permits to better isolate the signals from the aerosol-radiation-cloud interactions due to the so-called climate change penalty alone, while reduces the reliability of the future projections obtained, in fact. We now we make more emphasis on this point in the manuscript.

**- The last major concern is about the RCM used in this study. The version of WRF used here, namely 3.6.1 is quite old (reference paper from 2008), and above all a precise description of how aerosols and their effects on climate are represented is missing. For example, I wonder what aerosol climatology is used in the BASE simulation (if it is not zero). I am also very worried about the very low values of summer AOD shown in Figure 1g-h, which shows that WRF clearly underestimates AOD over Europe. WRF values range from 0.05 to 0.09 over**

**Europe, while observations typically range from 0.1 to 0.2 (Papadimas et al. 2008, Nabat et al. 2013, Schultze and Rockel 2018). That could lead to an underestimation of aerosol effects. In such a study, an evaluation of AOD (even brief) is needed in order to ensure the consistency of the results.**

Some of the simulations included in this work were performed time ago. Others have been performed more recently, but we decided not to change the WRF version to be sure of being comparing the same "thing". At that time, when first simulations were carried out, the last stable version of WRF was the 3.6.1. In any case, the physics of the model is the same, no matter of its version.

Addressing the appropriate reviewer concern about the lack of a deep description of how aerosols are treated by the model, we have now added more details on that. For instance, regarding the BASE experiments, we now say in section 2:

"BASE: aerosols are not considered in the simulations. No aerosol climatology is used and no aerosol interactions are taken into account by the model. WRF-alone considers a constant number of cloud condensation nuclei (250 per $cm^3$, set in the model by default) to enable the formation of clouds."

Regarding the ARCI experiments, we now explain in section 2:

"Aerosol-cloud interactions were implemented by linking the simulated cloud droplet number with the microphysics schemes (Chapman et al 2009) affecting both the calculated droplet mean radius and the cloud optical depth. Although this WRF-Chem version (3.6.1) does not allow a full coupling with aerosol-cloud interactions, the microphysics implemented here is a single moment scheme that turns into a two moments scheme in the simulations denoted as ARCI. One-moment microphysical schemes are unsuitable for assessing the aerosol-clouds interactions as they only predicts the mass of cloud droplets and does not represent the number or concentration of cloud droplets (Li et al. 2008). The prediction of two moments provides a more robust treatment of the particle size distributions, which is key for computing the microphysical process rates and cloud/precipitation evolution. In this sense, although the Lin microphysics is presented as a single moment scheme, the WRF-Chem model allows to transform the single into a double moment scheme. A prognostic treatment of cloud droplet number was added (Ghan et al. 1997), which treats water vapour and cloud water, rain, cloud ice, snow, and graupel. The autoconversion of cloud droplets to rain droplets depends on droplet number (Liu et al. 2005). Droplet-number nucleation and (complete) evaporation rates correspond to the aerosol activation and resuspension rates. Ice nuclei based on predicted particulates are not treated. However, ice clouds are included via the prescribed ice nuclei distribution following the Lin scheme. Finally, the interactions of clouds and incoming solar radiation have been implemented by linking simulated cloud droplet number with the Goddard shortwave radiation scheme, representing the first indirect effect, and with Lin microphysics, which represents the second indirect effect (Skamarock et al. 2008). Therefore, droplet number will affect both the calculated droplet mean radius and cloud optical depth."

References:

Ghan, S. J., Leung, L. R., Easter, R. C., & Abdul Razzak, H. (1997). Prediction of cloud droplet number in a general circulation model. *Journal of Geophysical Research: Atmospheres*, *102*(D18), 21777-21794.

Li, G., Wang, Y., & Zhang, R. (2008). Implementation of a two moment bulk microphysics scheme to the WRF model to investigate aerosol cloud interaction. *Journal of Geophysical Research: Atmospheres, 113*(D15).

Lin, Y. L., Farley, R. D., & Orville, H. D. (1983). Bulk parameterization of the snow field in a cloud model. Journal of Climate and Applied Meteorology, 22(6), 1065-1092.

Liu, Y., Daum, P. H., & McGraw, R. L. (2005). Size truncation effect, threshold behavior, and a new type of autoconversion parameterization. *Geophysical research letters, 32*(11).

Mitchell, D. L., Rasch, P., Ivanova, D., McFarquhar, G., & Nousiainen, T. (2008). Impact of small ice crystal assumptions on ice sedimentation rates in cirrus clouds and GCM simulations. *Geophysical research letters, 35*(9).

Skamarock, W.C.; Klemp, J.B.; Dudhia, J.; Gill, D.O.; Barker, D.M.; Wang, W.; Powers, J.G. A Description of the Advanced Research WRF Version 3; Technical Report NCAR Tech. Note TN-475+STR; NCAR: Boulder, CO, USA, 2008.

Tao, W. K., Simpson, J., & McCumber, M. (1989). An ice-water saturation adjustment. *Monthly Weather Review, 117*(1), 231-235.

And acknowledge in the discussion section:

"In the ARCI simulations, the autoconversion scheme called so that cloud droplets can turn into rain droplets is different to the autoconversion scheme activated in the ARI simulations. This change in the WRF-Chem configuration can lead to ARCI-ARI differences that do not come necessarily from the of the aerosol-cloud interactions from a physical point of view (Liu et al 2005). In fact, the activation of the aerosol-cloud interactions requires further changes in the model configuration (as compared to the configuration used for the simulations labeled as ARI) beyond the autoconversion scheme, such as the activation of aqueous chemistry processes, that could also have an added impact to effect that can be strictly attributed to the aerosol-cloud interactions. However, technically, the encoding of WRF-Chem model hampers to better isolate the effect of the aerosol-cloud interactions. Therefore, ARCI-ARI differences can not be attributed to the aerosol-cloud interactions from a purely physical point of view, but to the activation of the aerosol-cloud interactions from a modeling point of view, since the autoconversion schemes necessarily change between ARI and ARCI. This should be beared in mind when interpreting the signals."

Reference:

Liu, Y., Daum, P. H., & McGraw, R. L. (2005). Size truncation effect, threshold behavior, and a new type of autoconversion parameterization. *Geophysical research letters, 32*(11).

Regarding the inclusion of gas-phase chemistry in the simulations, in section 2 we added:

"Chemical reactions in the GOCART model include several oxidation processes by the three main oxidants in the troposphere: OH, NO3, and O3. The OH radical dominates oxidation during the daytime, but at night its concentration drops and NO3 becomes the primary oxidant (Archer-Nicholls et al., 2014). So, the oxidation pathways represented in GOCART include: (a) the dimethyl sulphide (DMS) oxidation by the hydroxyl radical (OH) during the day to form sulphur dioxide ($SO_2$) and methanesulfonic acid (MSA); (b) the oxidation by nitrate radicals (NO3) at night to form $SO_2$; and (c) the $SO_2$ oxidation by OH in air and by $H_2O_2$ and tropospheric ozone ($O_3$) in clouds (aqueous chemistry) to form sulphate (Chin et al., 2000). Henceforth, the skilful characterization of

gas-phase radicals such as OH and NO3 or compounds like $O_3$ is essential for the representation of oxidation pathways in the atmosphere leading to the formation of secondary aerosols (Jiménez et al., 2003). Therefore, in this contribution the RACM (Stockwell et al., 1997; Geiger et al., 2003) mechanism has been coupled to GOCART trough the kinetics pre-processor (KPP) in WRF-Chem in order to provide the concentrations of radical and gas-phase pollutants needed by the GOCART aerosol model."

References:

Archer-Nicholls, S., Lowe, D., Utembe, S., Allan, J., Zaveri, R. A., Fast, J. D., Hodnebrog, Ø., Denier van der Gon, H., McFiggans, G. (2014). Gaseous chemistry and aerosol mechanism developments for version 3.5.1 of the online regional model, WRF-Chem. *Geoscientific Model Development*, 7, 2557–2579.

Chin, M., Rood, R.B., Lin. S.-J., Müller, J.-F., Thompson, M. (2000). Atmospheric sulfur cycle simulated in the global model GOCART: Model description and global properties. *Journal of Geophysical Research*, 105(D20), 24671-24687.

Geiger, H., Barnes, I., Bejan, I., Benter, T., & Spittler, M. (2003). The tropospheric degradation of isoprene: an updated module for the regional atmospheric chemistry mechanism. *Atmospheric Environment*, 37, 1503 - 1519.

Jiménez, P., Baldasano, J.M., Dabdub, D. (2003). Comparison of photochemical mechanisms for air quality modeling. *Atmospheric Environment*, 37, 4179-4194.

Stockwell, W. R., Kirchner, F., Kuhn, M., & Seefeld, S. (1997). A new mechanism for regional atmospheric chemistry modeling. *Journal of Geophysical Research: Atmospheres*, 102, 25 847 - 25 879.

Finally, regarding the 'low' AOD values shown in Figure 1g-h, we must acknowledge that we certainly used wrong AOD values. These had been computed from the TAUAER3 and TAUAER4 variables, which do exhibit a weird evolution along the year. After inspection, we figured out that these and the EXTCOF55 variables had been wrongly recorded in the wrfout files (not new, apparently, see e.g.: https://forum.mmm.ucar.edu/phpBB3/viewtopic.php?t=9313&p=17464). So we have now adopted an alternative method to compute AOD following Palacios-Peña et al (2020), where, in fact, the representation of AOD by these model configurations (ARI and ARCI) were deeply evaluated. The new AOD files were estimated using the reconstructed mass-extinction method (Malm et al 1994) from the well-recorded concentrations of the various aerosol species in the wrfout files, namely: black carbon, organic carbon, dust and sea salt. Sulfates were estimated from SO2 and OH recorded concentrations using the same kinetic reaction as the one implemented in the RACM-KPP module. We want to remark that the mistake occured during the postprocessing of the wrfout files, while WRF-Chem run satisfactorily. These wrfout files were removed after postprocessed, so we have now generated a sample one (using the ARI configuration) and uploaded it for checking together with all the other data files. Importantly, this change in the methodology for estimating AOD values did not alter the the overall results of the paper. Attached here a figure with the AOD climatologies from ARI and ARCI and those from MACv2.

References:

Malm, W. C., Sisler, J. F., Huffman, D., Eldred, R. A., & Cahill, T. A. (1994). Spatial and seasonal trends in particle concentration and optical extinction in the United States. *Journal of Geophysical Research: Atmospheres*, 99(D1), 1347-1370.

Palacios-Peña, L., Montávez, J. P., López-Romero, J. M., Jerez, S., Gómez-Navarro, J. J., Lorente-Plazas, R., ... & Jiménez-Guerrero, P. (2020). Added Value of Aerosol-Cloud Interactions for Representing Aerosol Optical Depth in an Online Coupled Climate-Chemistry Model over Europe. Atmosphere, 11(4), 360.

**Other comments:**

**- page 2 line 31: land use change is not specific to regional climate simulations, I think it is even more used in global climate simulations.**

We simply intend to note that at higher resolutions, land uses can be represented at a finer scale.

**- page 3 lines 57-63: please avoid such long lists of references, and clarify the conclusions of each of them.**

This paragraph has been reformulated accordingly.

**- page 3 lines 70-71: "which still remain largely a mystery". Other studies such as Giorgi et al. (2016), Sørland et al. (2018) and Boé et al. (2020) have also underlined differences between RCMs and GCMs in future projections. The role of aerosols is even discussed in Boé et al. (2020), which should be mentioned here.**

In fact. This is now acknowledged in the paper.

**- page 5 lines 140-141: it is not clear for me how aerosol-cloud interactions are represented in the simulations.**

We now provide further details in the manuscript. See our response about this above.

**-page 6 lines 144-146: This way of calculating clear-sky variables in simulations is not common in modeling studies. It would be appropriate for a comparison to observations (it is exactly how satellites do for example), but in models, you generally compute clear-sky variables at each time step, removing clouds in radiative transfer. This would avoid the numerous missing values.**

Unfortunately, we did not save clear-sky values from the model outputs, so we needed to adopt an alternative methodology.

**- page 6 section 3: This section should be divided in several sub-sections, with more precise titles than only "Results".**

Done.

**- page 6 line 165: "The inclusion of interactive aerosols reduce the JJA mean values of RSDS". This is typically an example of my first main comment. This decrease in rsds is likely due to the mean effect of aerosols, and not their interactive pattern.**

We agree that the sentence was misleading and has been reformulated. Although that statement is true in the context of our work, it wrongly gave the message that such a reduction in RSDS is due to the interactive aerosols modeling approach adopted here as compared to a more conservative (and common) approach based on non-interactive aerosols, which is something that we did not inspect.

**- page 7 line 172: "ARI and ACI lead to more cloudiness in central and northern regions". This is not really the case when looking at the figure.**

Blue colors prevail in central and northern regions in Figure 1d-f indeed. Please note that the color palette is inverted here with the aim of facilitating a visual identification of the matching between the patterns of the drivers (CCT, AOD) and those of RSDS.

**- page 7 lines 184-187: This conclusion is not justified.**

Both the spatial correlations shown in the previous Supp. Figure 4 and the temporal correlations shown in the previous Supp. Figure 5 (Supp. Figure numbers changed in the new version) support that the CTT differences prevail over the AOD differences in driving the RSDS differences between pairs of experiments, not only in the present-day climate simulations, but also under future conditions.

**- Figures 1-4: From my point of view it would be easier to understand to have differences in absolute values rather than in percentages. Indeed, I suspect here we look at very low values which could be unsignificant.**

Figures 1 to 4 have been replicated to show plain differences. These new figures have been included as Supp. Material and used to describe the results.

**- Figures 1-4: Why consider only land points ? It would be interesting to show also ocean points on figures.**

We prefer to focus only on land points to get a clearer message tailored to the modeling applications for the solar energy sector.

**- Figure 3: When comparing the evolution of rsds, cct and aod in the simulations, I suspect a possible bug in the figure or in the simulation. Indeed, the strong decrease in rsds in northern latitudes (for example in Iceland), is neither explained by cct nor by aod.**

The negative signals in rsds must be related to the increase in cct, even if small. Maybe better to see the plots with the differences in absolute values provided in the new version of the Supp. Material. This is also supported by the high spatial correlations between rsds and cct changes.

**- Page 8 lines 218-219: If "the anthropogenic component is disregarded", there should be no possible conclusion on the future evolution of rsds.**

We do not attempt to provide reliable projections for RSDS, but to unveil how aerosols can affect them. See our response to the main comment #2.

**- The manuscript suffers from many typographical and English spelling errors that need to be corrected.**

The manuscript has been entirely revised with the help of a native speaker.

**AOD climatologies for 1991-2010**

[Figure]

**(a)** DJF MACv2  **(b)** MAM MACv2  **(c)** JJA MACv2  **(d)** SON MACv2

**(e)** DJF ARI  **(f)** MAM ARI  **(g)** JJA ARI  **(h)** SON ARI

**(i)** DJF ARCI  **(j)** MAM ARCI  **(k)** JJA ARCI  **(l)** SON ARCI

0.00   0.05   0.10   0.15   0.20   0.25

---

## Author Comment (AC4) · 15 Oct 2020

Authors' response

**This manuscript, submitted to Geoscientific Model Development, aims to identify the role of interactively modeling aerosol in regional climate simulations over Europe, by conducting a sensitivity study with the WRF model. The focus is on solar radiation at the surface during summer. Both a present and a future period are considered. Changes in cloudiness are presented as the main driver of the changes in solar radiation. There are some interesting features in this study, such as long simulations with the WRF-Chem model using interactive aerosol that are computationally demanding. However, I believe that the main problem is that the aim of the study is not actually addressed. I believe that separating the "interactive" part of aerosol modeling and making general comments about it is not possible in the current study. Thus it is a problem of methodology and structuring of the whole manuscript. Moreover I believe that a significant clarification is need in the current methodology regarding the BASE simulation that is the basis for comparison. I would hesitate to recommend it for publication in its current form. However, I believe that it could stand as a sensitivity study aiming to describe the impact of the specific model and aerosol treatments used. I would suggest major revisions regarding: the aims of the study, including a validation, possibly changing the analysis under clear-sky conditions, clarifying the aerosol treatment in the simulations. In the end, I think the study could provide some interesting points to the community.**

We do thank the reviewer for the time devoted to read and thoughtfully comment on our work. Below we provide detailed answers to each comment, hoping to have been clear enough in our explanations. Attending these comments and the ones posted by the other reviewers and the Editor, the new version of the manuscript:

1 – Has been entirely revised by a native speaker in order to improve the redaction.

2 – Has a new title. The change intends to avoid that the reader interprets that we are comparing simulations with dynamic vs. static aerosols. The new title is:

"Sensitivity of surface solar radiation to aerosol-radiation and aerosol-cloud interactions over Europe in WRFv3.6.1 climatic runs with fully interactive aerosols"

In this line, we have made an effort to make the scientific purpose of the manuscript clearer throughout the whole text.

3 – Includes further details and arguments on the experimental set-up and the methodology. Section 2 has been divided into 3 subsections.

4 – The formerly labeled as ACI simulations are now named ARCI to emphasize that these include both aerosol-radiation and aerosol-clouds interactions.

5 – Includes a brief validation exercise. We now face the outputs of our simulations with ERA5.

6 – Includes two subsections within the Results section: one for the historical simulations and another for the future projections.

7 – Includes a deeper discussion of the results attending several comments by the reviewers, e.g.:

- the activation of the autoconversion scheme in the ACI simulations hampers a direct attribution of the signals to the aerosol-cloud interactions from a physical point of view (the attribution can be made, from a modeling point of view, to the activation of these interactions in the model);

- the fact that we kept constant the anthropogenic aerosol emissions in future simulations permits to better isolate the signals from the aerosol-radiation-cloud interactions due to the so-called climate change penalty alone, while reduces the reliability of the future projections obtained;

- the signals obtained for different seasons (additional analysis is provided as Supplementary Material);

8 – Includes a link where all the data and codes to reproduce our study have been made publicly available: http://doi.org/10.23728/b2share.682b1c6311134b36a18f59a99a443afd.

We are confident that these major changes have improved significantly the manuscript and provides a larger support to its key findings.

We must also notice that we used wrong AOD values in the previous version of the paper, as it was noted by the reviewer2. These had been computed from the TAUAER3 and TAUAER4 variables, which do exhibit a weird evolution along the year. After inspection, we figured out that these and the EXTCOF55 variables had been wrongly recorded in the wrfout files (not new, apparently, see e.g.: https://forum.mmm.ucar.edu/phpBB3/viewtopic.php?t=9313&p=17464). So we have now adopted an alternative method to compute AOD following Palacios-Peña et al (2020), where, in fact, the representation of AOD by these model configurations (ARI and ARCI) were deeply evaluated. The new AOD files were estimated using the reconstructed mass-extinction method (Malm et al 1994) from the well-recorded concentrations of the various aerosol species in the wrfout files, namely: black carbon, organic carbon, dust and sea salt. Sulfates were estimated from SO2 and OH recorded concentrations using the same kinetic reaction as the one implemented in the RACM-KPP module. We want to remark that the mistake occurered during the postprocessing of the wrfout files, while WRF-Chem run satisfactorily. These wrfout files were removed after postprocessed, so we have now generated a sample one (using the ARI configuration) and uploaded it for checking together with all the other data files. Importantly, this change in the methodology for estimating AOD values did not alter the the overall results of the paper.

**Major comments:**

**1. One of my major concerns is that the nature of the BASE experiment is not clear to me. It is stated that it works with a specific aerosol concentration and that "the aerosol radiative effect is assumed to come as an external forcing." I am not sure what this means. Does the BASE experiment let these aerosols interact with radiation? In this case the AOD field needs to be shown. Or their only impact is that they are just used by the microphysics to facilitate cloud formation? In any case, the nature of aerosol in the BASE experiment needs to be clearly stated so that the reader understands the results of the comparison. Moreover if BASE has an AOD that interacts with radiation, how much does it differ from the AOD of ARI and ACI? Are the differences between BASE and these simulations attributed to the difference in AOD and not to the introduction of dynamic aerosol?**

We agree that the BASE experiment was poorly described. Now we say:

"BASE: aerosols are not considered in the simulations. No aerosol climatology is used and no aerosol interactions are taken into account by the model. WRF-alone considers a constant number of cloud condensation nuclei (250 per $cm^3$, set in the model by default) to enable the formation of clouds."

**2. It is very interesting to try and identify the impact of interactively modeled aerosols. However, I am not sure that this is achieved in the study. You can make a statement that, for example, the ARI experiment that uses "this specific" interactive aerosol treatment in WRF-Chem has "this specific impact" on radiation. This statement could be useful to the community as a sensitivity study of the model and aerosol scheme. However, I do not think that you can attribute this impact only to the "interactive" part. Probably, a first step towards that direction would be to have additional experiments enabling aerosol-radiation and cloud interactions using static aerosol fields with the same mean AOD as the ones in ARI and ACI.**

We also agree here. We were wrongly giving the message that signals were due to the interactive aerosols modeling approach adopted here as compared to a more conservative (and common) approach based on non-interactive aerosols, which is something that we did not inspect. We have accordingly reformulated the title and redaction of the manuscript.

**3. I believe a validation (even a quick one) of the simulations, especially regarding rsds and AOD, should be part of the study in order to assert that they do capture the basic patterns of the examined variables. I do understand that they are compared against the GCM (and that the GCM has been probably validated), but still a validation would make the results more robust.**

The manuscript now includes a brief comparison of the present-day simulations with ERA5 (for RSDS). The representation of AOD by these model configurations (ARI and ARCI) were deeply evaluated in Palacios-Peña et al. (2020). Nonetheless, find attached here a figure with the AOD climatologies from ARI and ARCI and those from MACv2.

**4. The methodology to calculate Clear sky conditions was a bit unusual to me. I am aware that the radiation code in WRF (and I think this is the case for version 3.6.1) provides the clear-sky radiation at every time step simultaneously with rsds. It would probably be better to use that feature. I also have a question regarding the methodology. It is stated (page 6, 150-152) that in order to consider a specific grid point in the analysis you need to have at least 15 records per period that are not missing values. Ok so far. It is stated (page 6, lines 153-154) that "(which, according to our methodology, would occur only if all days within a summer season have CTT values >1%)." So, if I understand correctly even if one day within a summer season has a CCT value <1, that summer season gains a valid value based only on that day and is considered in the analysis?**

Unfortunately, we did not save clear-sky values from the model outputs, so we needed to adopt an alternative methodology.

Regarding the methodology, the reviewer's interpretation is right. In any case, we show seasonal climatological values (or differences), thus the results are independent of that (we simply average over all days with CCT<1%). The number of seasons with non-missing values just affects the interannual variability of the seasonal series, thus playing a role when assessing the statistical significance of the differences between that climatological values. Therefore, outliers values (in case) should affect very little the overall results.

**5. The use of no time evolving anthropogenic aerosol in the future period by ARI and ACI experiments is not ideal. It is good that this deficiency is stated in the manuscript (page 8, line 218). Moreover, it would be interesting to see what are the rsds differences between the GCM and ARI/ACI for the future period.**

As stated above, this approach permits to better isolate the signals from the aerosol-radiation-cloud interactions due to the so-called climate change penalty alone, while reduces the reliability of the future projections obtained, in fact. We now we make more emphasis on this point in the manuscript.

Attached a figure with ARI-GCM and ARCI-GCM differences in rsds in the future period. Not that different to those in the present period.

**Minor comments:**

**-Page 1, line 20 "reduction about 5% in RSDS was found when aerosols are dynamically solved". This is compared to BASE? It must be clearly stated.**

Done.

**-Page 2, line 33 The phrase "all about cumulus" I believe should be clarified a bit better. Is this about convective phenomena, the cloud fraction scheme or both?**

It is more correct to say "convective phenomena" indeed. Amended.

**-Page 4 lines 97-98. In the BASE experiment "the by-default WRF setup was used, which considers 250 cloud condensation nuclei per cm3 to form clouds". I think the term "by-defalut" might be a bit misleading. I understand that this concentration of CCN is probably related to the Lin microphysics scheme used in the experiments and this should be stated.**

This part has been reformulated. Anyway, this CCN value (250 per cm3) is not linked to the microphysics scheme, but something more general in the model.

**-I do not understand how ACI (page 5, lines139-141) works. What is meant by "Although this WRF-Chem version (3.6.1) does not allow a full coupling with aerosol-cloud interactions. . ."? I believe it should be clearly stated which are the parts of the aerosol-clouds interactions that are missing. Also I think it should be stated to which variables the single and double moment treatment is applied.**

We have amended the lack of description of the WRF setup used to perform the simulations labeled as ARCI by including the follwoing in the text (in section 2):

"Aerosol-cloud interactions were implemented by linking the simulated cloud droplet number with the microphysics schemes (Chapman et al 2009) affecting both the calculated droplet mean radius and the cloud optical depth. Although this WRF-Chem version (3.6.1) does not allow a full coupling with aerosol-cloud interactions, the microphysics implemented here is a single moment scheme that turns into a two moments scheme in the simulations denoted as ARCI. One-moment microphysical schemes are unsuitable for assessing the aerosol-clouds interactions as they only predicts the mass of cloud droplets and does not represent the number or concentration of cloud droplets (Li et al. 2008). The prediction of two moments provides a more robust treatment of the particle size distributions, which is key for computing the microphysical process rates and cloud/precipitation evolution. In this sense, although the Lin microphysics is presented as a single moment scheme, the

WRF-Chem model allows to transform the single into a double moment scheme. A prognostic treatment of cloud droplet number was added (Ghan et al. 1997), which treats water vapour and cloud water, rain, cloud ice, snow, and graupel. The autoconversion of cloud droplets to rain droplets depends on droplet number (Liu et al. 2005). Droplet-number nucleation and (complete) evaporation rates correspond to the aerosol activation and resuspension rates. Ice nuclei based on predicted particulates are not treated. However, ice clouds are included via the prescribed ice nuclei distribution following the Lin scheme. Finally, the interactions of clouds and incoming solar radiation have been implemented by linking simulated cloud droplet number with the Goddard shortwave radiation scheme, representing the first indirect effect, and with Lin microphysics, which represents the second indirect effect (Skamarock et al. 2008). Therefore, droplet number will affect both the calculated droplet mean radius and cloud optical depth."

References:

Ghan, S. J., Leung, L. R., Easter, R. C., & Abdul Razzak, H. (1997). Prediction of cloud droplet number in a general circulation model. *Journal of Geophysical Research: Atmospheres*, *102*(D18), 21777-21794.

Li, G., Wang, Y., & Zhang, R. (2008). Implementation of a two moment bulk microphysics scheme to the WRF model to investigate aerosol cloud interaction. *Journal of Geophysical Research: Atmospheres*, *113*(D15).

Lin, Y. L., Farley, R. D., & Orville, H. D. (1983). Bulk parameterization of the snow field in a cloud model. Journal of Climate and Applied Meteorology, 22(6), 1065-1092.

Liu, Y., Daum, P. H., & McGraw, R. L. (2005). Size truncation effect, threshold behavior, and a new type of autoconversion parameterization. *Geophysical research letters*, *32*(11).

Mitchell, D. L., Rasch, P., Ivanova, D., McFarquhar, G., & Nousiainen, T. (2008). Impact of small ice crystal assumptions on ice sedimentation rates in cirrus clouds and GCM simulations. *Geophysical research letters*, *35*(9).

Skamarock, W.C.; Klemp, J.B.; Dudhia, J.; Gill, D.O.; Barker, D.M.; Wang, W.; Powers, J.G. A Description of the Advanced Research WRF Version 3; Technical Report NCAR Tech. Note TN-475+STR; NCAR: Boulder, CO, USA, 2008.

Tao, W. K., Simpson, J., & McCumber, M. (1989). An ice-water saturation adjustment. *Monthly Weather Review*, *117*(1), 231-235.

We have also acknowledged in Discussion the following:

"In the ARCI simulations, the autoconversion scheme called so that cloud droplets can turn into rain droplets is different to the autoconversion scheme activated in the ARI simulations. This change in the WRF-Chem configuration can lead to ARCI-ARI differences that do not come necessarily from the of the aerosol-cloud interactions from a physical point of view (Liu et al 2005). In fact, the activation of the aerosol-cloud interactions requires further changes in the model configuration (as compared to the configuration used for the simulations labeled as ARI) beyond the autoconversion scheme, such as the activation of aqueous chemistry processes, that could also have an added impact to effect that can be strictly attributed to the aerosol-cloud interactions. However, technically, the encoding of WRF-Chem model hampers to better isolate the effect of the aerosol-cloud interactions. Therefore, ARCI-ARI differences can not be attributed to the aerosol-cloud interactions from a purely physical point of view, but to the activation of the aerosol-cloud

interactions from a modeling point of view, since the autoconversion schemes necessarily change between ARI and ARCI. This should be beared in mind when interpreting the signals."

Reference:

Liu, Y., Daum, P. H., & McGraw, R. L. (2005). Size truncation effect, threshold behavior, and a new type of autoconversion parameterization. *Geophysical research letters*, *32*(11).

**-I believe it is useful to know which statistical test is used (t-set, non parametric Mann-Whitney. . .) to determine statistical significance.**

We used the t-test. Section 2 has now been splitted into several subsections. The last one includes more methodological details, as this one.

**-Total cloud cover values over southern Europe in summer are usually small. Thus, the changes in CCT between the experiments could be in some cases negligible but the relative (percentage) change could be inflated. I believe this should be stated in the manuscript. Also, it would be interesting to see a plot with the plain difference in CCT between experiments in the supplement.**

Figures 1 to 4 have been replicated to show plain differences. These new figures have been included as Supp. Material and used to describe the results.

**-Page 7, lines 185-186. "Contrary, the effect of interactive aerosols schemes. . ." The way it is written gives the impression that the authors are talking about interactive schemes in general. I think it would be better to avoid generalizing the results of this specific sensitivity study.**

The reviewer is right. We have followed this suggestion all along the revised manuscript.

**-Page 8, lines 209-210. "These latter are more widespread in ARI than in BASE, which makes the ARI pattern the most similar to the change pattern from the GCM". I do not clearly see this in Figure3.**

The ARI pattern (Fig 3 c) shows the most widespreaded positive signals south-eastward, and the lowest negative signals northward.

**Technical corrections:**

**Page 7 line 183 "varables" -> variables**

**Page 7, line 188 I am not aware of the word "devanishes". Could this be a spelling mistake?**

**Page 10, line 274 experimts -> experiments**

**Page 1, line25 much more softer -> much softer**

Thanks for these corrections. The entire manuscript has been revised by a native speaker.

**AOD climatologies for 1991-2010**

[Figure]

**(a)** DJF MACv2  **(b)** MAM MACv2  **(c)** JJA MACv2  **(d)** SON MACv2

**(e)** DJF ARI  **(f)** MAM ARI  **(g)** JJA ARI  **(h)** SON ARI

**(i)** DJF ARCI  **(j)** MAM ARCI  **(k)** JJA ARCI  **(l)** SON ARCI

0.00   0.05   0.10   0.15   0.20   0.25

**RSDS JJA climatologies for 2031-2050**

[Figure]

RSDS summer climatologies in the future period from the GCM (a) and the WRF simulations (b to d); units: W/m2. Panels e to g depict relative differences between each WRF simulation and the GCM, squared if statistically significant (p<0.05); units: %.

---

## Referee Report (RR1)

**Review of the manuscript entitled** *Sensitivity of surface solar radiation to aerosol-radiation and aerosol-cloud interactions over Europe in WRFv3.6.1 climatic runs with fully interactive aerosols,* **by S. Jerez et al.**

This is the revised version of the manuscript, submitted to *Geoscientific Model Development*, which presents a sensitivity study on the role of dynamic aerosols in regional climate simulations over Europe, carried out with the WRF model.

The authors have made substantial efforts to improve the manuscript, and to take into account the different suggestions of the reviewers. The objective of the paper and the results are clarified and better presented. English spelling has also been improved. However, before the publication in GMD, I recommend the following corrections.

**Main comments:**

- The conclusions about the prevailing of aerosol indirect effects over direct effects should be moderated, since they could be very model-dependent. They could also depend on the choice of the parameterization of cloud-aerosol interactions. More discussions about this aspect should be added in the text.

- Many figures are in supplementary material, and are often used in the text of the manuscript. Some sections entirely rely on supplementary figures. At the end, the revised version has only 4 figures in the main text. I think that more figures (not all obviously) should be included in the main text rather than in the supplementary. These figures are essential to better understand the study.

- Section 3.1 : The brief evaluation of AOD (as shown in Figure page 21 in your replies) should be added to the manuscript (at least for JJA). It is an important point to understand the rest of the study. It would come in addition to the brief validation of RSDS in the beginning of section 3.1 (line 229).

- Sections 3.1 and 3.2 could be separated in different subsections in order to avoid too long paragraphs, and to make reading more fluent.

- The authors have mentioned in their replies that they have included in the discussions the justification of keeping constant emissions in future simulations to evaluate the climate change penalty. However I did not find that in the revised version.

**Other comments:**

- page 1 line 12: please avoid undefined acronyms (IPCC) in the abstract

- page 1 line 19: Since the period of simulations is 1991-2010, "historical period" would be better than "present-day conditions" (1991 is 30 years ago!). In the whole paper, present could be replaced by historic (or past).

- page 5 line 121: Is the RRTM radiative scheme used both for shortwave and longwave or longwave only? Please mention the radiative scheme for shortwave in the second option.

- page 5 line 132: please use subscript characters for chemical species such as $NO_3$ and $O_3$

- page 5 line 142: through instead of trough

- page 6 line 148: "Anthropogenic emissions … were kept unchanged in 148 the simulation periods (we considered the 2010 monthly values)". Anthropogenic emissions have been dramatically

reduced between the 1980s and 2010, so keeping 2010 values and comparing simulated aerosols over the whole 1991-2010 period could lead to an underestimation of AOD.

- page 6 line 155: "aerosol optical properties assuming wet particle diameters". Which humidity is used for the calcuation of aerosol optical properties? Is the variation of humidity taken into account on-line in the simulation to allow variations of optical properties (through pre-defined intervals for example) ?

- page 8 line 216: Could you give the level of significance for the t-test ?

- page 13 line 389-390: This point about the autoconversion scheme seems to me very important. Could you explain why you could not use the same scheme, but for example with constant vallue of autoconversion in the case without aerosols ?

---

## Author Response (AR2)

Authors' response

**The restructuring of the paper's theme has improved the readability and scope dramatically. I now understand how the different model runs are important to present to the scientific community and falls under the scope of GMD.**

We are glad to hear that, and sincerely thanks the reviewer for agreed to revise again our manuscript, especially during this hard times we all are living, and for the constructive comments provided.

**Overall Comments:**

**-The reversal of the colorbar between the plots is confusing (sometimes red is an increase and sometimes it's a decrease). Is there a reason for this? Typically, readers associate blue with a reduction and red with an increase. It would help if it was consistent throughout the paper.**

Some of us think the same indeed. The reversal of the colorbars for CCT and AOD (and others) intends to facilitate a visual matching (reds-reds, blues-blues) between the RSDS plots and those of their inspected drivers (in particular, CCT and AOD), since one expects that a reduction in CCT (or AOD) leads to increased RSDS. So, where red colors appear in the RSDS plots, indicating an increase, red colors are expected in the CCT (or AOD) plots, indicating a reduction in this case because the colorbars are reverted.

**-It may help if some of the supplementary figures are in the paper as official figures. It is hard to flip back and forth, plus the supplementary figures are referenced more than the included plots. The correlations could state supplementary, but the aerosol type and precipitation / environment plots are important and should be included in the manuscript.**

Following this suggestion, also indicated by the other reviewer, we have now moved the following figures, previously in Supplementary Material, to the main manuscript:

Previously Supp Fig 4 (top panels), now Fig 2: Contribution of each aerosol species to the JJA-mean total surface aerosol concentration in the period 1991-2010. This is now Fig 2 in the revised manuscript.

Previously Supp Fig 5, now Fig 3: Differences between experiments in the RSOT, TAS, RH & CLD JJA climatologies for 1991-2010.  This is now Fig 3 in the revised manuscript.

Previously Supp Fig 6, now Fig 5: Differences between experiments in precipitation-related JJA climatologies for 1991-2010. This is now Fig 4 in the revised manuscript.

**Specific Comments:**

**[39-40] "However, there are relevant processes that GCMs usually model dynamically, but which RCMs usually do not" – my understanding is that GCMs parameterize and specify constant values more than an RCM because RCMs have better resolution? Should GCM and RCM be switched in this sentence? I don't see how WRF would model fewer things dynamically than a GCM, especially related to aerosol-cloud processes. Please correct my ignorance here if I'm reading this wrong.**

The sentence is certainly misleading, so we have reformulated as follows:

"However, there are relevant processes that GCMs usually model dynamically, but which are not usually included in RCMs runs."

It is not that RCMs model fewer things than GCMs, but that RCMs are usually run without a dynamical modeling of *things* such as aerosols, in contrast to the runs that are usually performed with GCMs. This increases the interest of our study. Check, for instance, the RCMs setups regarding aerosols that are being used under the Euro-Cordex umbrella in Gutiérrez et al. (2020). For our surprise, something similar happened with the (non-)inclusion of the time-evolving GHG concentrations in climatic runs performed with RCMs (Jerez et al. 2018).

**[43-45] - There are also longwave and shortwave absorbing aerosol types like dust and smoke that can warm aerosol layers aloft while decreasing surface temperatures, leading to stable layers. This sounds like it's being discussed right after the part on semi-direct effects, but it is a direct effect. The citations are clumped together for these two things when they are distinct.**

The reviewer is right in this appreciation. There are different paths for clouds inhibition due to the warming effect of aerosols by absorption of solar radiation: (1) by heating the air and thus reducing the relative humidity, (2) by heating the air and thus provoking the evaporation of clouds (clouds burn-off), and (3) by leading to more stable atmospheric situations (as pointed out by the reviewer). 1 and 2 would be semi-direct, but 3 direct, according to the reviewer appreciation and to the IPCC definition. So we have reformulated as follows:

"Depending on their nature and the ambient conditions, aerosols can act to scatter and/or absorb the solar radiation through ARI, which may result in less or more solar radiation reaching the surface through direct and semi-direct effects. Direct effects might involve that less solar radiation reaches the surface due to its scattering and absorption (Giorgi et al 2002, Nabat et al 2015a, Li et al 2017, Kinne 2019), or more if, for instance, absorption warms aloft atmospheric layers, thereby leading to more stable atmospheric situations (lower surface temperatures than upward) and thus to the inhibition of clouds formation via convective phenomena (Giorgi et al 2002, Nabat et al 2015a). Absorption itself can also lead to clouds inhibition and/or burn-off through thermodynamic effects, i.e. by heating the air (semi-direct effects), thus increasing the amount of solar radiation reaching the surface (Allen and Sherwood 2010)."

**[144] - The RACM-GOCART combination produces SOA, but it may only couple to WRF in the form of radiative effects. I don't know that SOA is connected as an indirect effect. Please check this.**

The reviewer is right, SOA is not connected as an indirect effect. We did not say that, but just that RACM is used to provide GOCART the concentrations of radical and gas-phase pollutants (which may form SOA) that it needs.

**[159-160] - WRF-Chem does have fully coupled aerosol-cloud-radiation modules, including MADE and MOSAIC, they just were not selected here most likely because they are computationally expensive.**

From a climatic point of view, the aerosols exerting the highest influence in climate are sea salt and desert dust. MADE and MOSAIC aerosol schemes do not include the aforementioned types of aerosols in their formulation, so the use of GOCART is not only because of its cheaper computational time, but because of the need of taking into account dust and sea salt aerosols in our simulations. The limitations of MADE and MOSAIC are negligible for specific episode

formulations where the influence of natural aerosols might be negligible, but that is not the case for climatic simulations. We have added this clarification in the manuscript.

**[165-167] - "The WRF-Chem model makes it possible to transform the single- into a double-moment scheme" – this sentence is misleading and needs to be phrased more like "The WRF-Chem model assumes XXX to infer an aerosol number concentration from aerosol mass" and the parts about converting from single to double moment should be removed.**

The sentence follows the title of the publication by Li et al. (2008): "Implementation of a two moment bulk microphysics scheme to the WRF model to investigate aerosol cloud interaction."

Nonetheless, we have reformulated, to expand the description of the implemented mechanisms, as follows:

"(...) although the Lin microphysics is originally presented as a single moment scheme (Lin et al 1983), a modified Lin double-moment microphysical scheme is implemented in WRF-Chem (Lin et al 2008) and used here to conduct the ARCI simulations. In this scheme, both the mass and the total number of cloud droplets are predicted. The prognostic treatment of cloud droplet number involves water vapor, cloud water, rain, cloud ice, snow and graupel (Ghan et al 1997), and is activated through the "mixactivate" module of WRF-Chem. In that module, WRF-Chem calculates the aerosol number per volume concentration by using, for each aerosol type, the information about the size (the mean volume-diameter of each aerosol mode, obtained from the aerosol mechanism implemented in the simulation), and fixed densities and molecular weight of each type of aerosols. With all this information and the total mass, WRF-Chem estimates the aerosol number for each mode assuming spherical particles. The autoconversion of cloud droplets to rain droplets depends on droplet number (Liu et al 2005). Droplet-number nucleation and (complete) evaporation rates correspond to the aerosol activation and resuspension rates. Ice nuclei based on predicted particulates are not treated. However, ice clouds are included via the prescribed ice nuclei distribution, following the Lin et al (2008) scheme. Thus, the droplet number will affect both the calculated droplet mean radius and cloud optical depth. Finally, the interactions of clouds and incoming solar radiation were implemented by linking the simulated cloud droplet number with the Goddard shortwave radiation scheme, representing the first indirect effect (i.e. increase in droplet number associated with increases in aerosols), and with the Lin microphysics, representing the second indirect effect (i.e. decrease in precipitation efficiency associated with increases in aerosols)."

**I think my point in the first round of comments was missed. The Ghan et al. (1997) CCN formulas still require an input of aerosol number. GOCART and Lin microphysics does not predict aerosol number because it's single moment. Thus, some assumption must be made to get at aerosol number with this setup. This conversion between mass and number doesn't magically transform a single moment scheme into a double moment scheme because there is no value-added information: it is an assumption to make the model work. I'm asking the authors to check in the code or reach out to the WRF-Chem team to understand how this setup gets number information from mass.**

**I urge the authors to not use the model as a black box and understand what is being represented, what is being assumed, and what is not in the model whatsoever. The authors may think I'm being unreasonable here, but this is important for understanding the limitations of this study and to inform researchers who may look to this setup for future work.**

We perfectly understand the reviewer and have followed his/her suggestions in this sense. We have expanded the description of how the model works under the chosen setup to make the limitations of this study clearer.

**[169] – The calling of a different autoconversion scheme should be mentioned in the model setup section too and not just at the end. It's good to know limitations before looking at the data.**

Done, we have added the following in section 2.2:

"An important aspect of the differences in the model setup between experiments is that the autoconversion scheme necessarily changes in the ARCI simulations as compared to the model configuration used for ARI and BASE. The flag *progn* of the WRF namelist should be set to 0 for running ARI experiments in order to keep disabled the interaction of the online-estimated aerosols with cloud microphysics, hence ensuring the use of prescribed aerosols (as in the case of the BASE simulations) as this regards. Conversely, *progn* should be set to 1 for running ARCI experiments in order to feed the cloud microphysics scheme with the online-estimated number and physico-chemical properties of aerosols (this effectively turns the Lin scheme into a second-moment microphysical scheme)."

**[247-248] "With higher aerosol concentrations over most of the domain, reducing it by up to a half" – does this mean that where there are higher aerosol concentrations, the reduction is stronger?**

No, that was not the sense of the sentence, which was probably poorly redacted. We have reformulated as:

"the indirect aerosols effects tend to counteract the joint direct and semi-direct effects seen in the ARI minus BASE pattern, reducing it by up to a half over most of the domain"

**[259] – More stratiform clouds? More convective clouds? Because lower surface temperatures, despite an increase in RH, can lead to less convection.**

Effectively, so we have reformulated as follows (see also the new Fig 4):

"Compared to BASE, both ARI and ARCI lead to more cloudiness in central and northern regions (albeit quite slight increases, well below 5%). This could respond to the direct effect of the scattering of the solar radiation due to the high presence of sea salt, dust and sulfate over these areas (Fig 2), as an increase in RSOT over these areas is also appreciated in both ARI and ARCI simulations (Fig 3a-b). In addition, this direct effect could be triggering the following feedback mechanism: the cooling effect downward (where less solar radiation is received because of its scattering) cools down surface temperatures (Fig 3d-e), thus increasing relative humidity (Fig 3g-h), which may favor the formation of clouds (these should be non-convective, mostly low-level, clouds as the decrease in TAS leads to more stable atmospheric layers; Fig 4a,b), thus less radiation reaches the surface, thus lower surface temperatures, and so on. Noteworthy, both the reduction in RSDS and the accompanying increase in RSOT is more marked in ARI than in ARCI over central regions (Fig 1c and Fig 3c), where the indirect effects included in the ARCI simulation, such as in-cloud aerosol scavenging processes, could lead to cleaner atmospheres than ARI simulates."

**[260-262] – The semi-direct effect would be in both the ACR and ACIR simulations, correct? This also applies to discussions of the semi-direct effect and it's attribution throughout the paper.**

Correct, both ARI and ARCI includes the semi-direct effect. The alluded sentence was wrong and has been removed (probably it was retained by mistake from intermediate versions of the article during the revision process). We revised all the attributions made in this sense to the aerosol effects along the manuscript.

**[269] - Large aerosols, or GCCN, can accelerate cloud processes such as nucleation and collision-coalescence. What do you mean by large aerosols would prevent cloud formation?**

Although wrongly expressed, we meant the same thing as the reviewer says in the last part of his/her comment regarding the acceleration of collision-coalescence processes: that large aerosols, by favoring the conversion of cloud droplets into rain droplets, "hamper the formation of clouds" because they fasten that precipitation occurs and thus clouds disappear. So we have reformulated as:

"(…) a high presence of large aerosols over southern Europe, both in form of dust or sulfate in our case (Fig 2), can accelerate collision-coalescence processes fastening that precipitation occurs and thus shortening the lifetime of clouds (...)"

**[278] – Could it be that because most of the aerosol mass is dust, that the absorption is creating stable layers and preventing convection?**

Could be, but we did not find a clear evidence for that. We have reformulated, inspecting new hypothesis, as follows (see also the new Fig 4):

"both ARI and ARCI lead to less cloudiness southward as compared to BASE, especially ARCI (reductions up to 10% in Mediterranean regions; Fig 1d-e). Consistently, the ARCI minus ARI pattern (Fig 1f) depicts negative values (around 5%) along the Mediterranean strip. Therefore, both semi-direct and indirect aerosol effects would tend to diminish cloudiness southward, with the latter (indirect effect) having the greatest impact. This could be due to the fact that a high presence of large aerosols over southern Europe, both in form of dust or sulfate in our case (Fig 2), can accelerate collision-coalescence processes fastening that precipitation occurs and thus shortening the lifetime of clouds (Lee et al 2008), which is most plausible in the warm season over warm areas (Yin et al 2000), as long as aerosol-cloud interactions are resolved by the model. However, we did not find such an enhanced precipitation effect in our simulations (maybe the signal does not hold at the climatic scales assessed here), only a decrease in both mean cloudiness and number of cloudy days (Supp Fig 3j-l) together with consistent pictures of lower mean precipitation, lower mean convective precipitation, fewer rainy days and lower extreme precipitation values emerging over those areas where the aerosol effects diminish cloudiness (Fig 5). The reduction in convective precipitation (the prevailing form of precipitation over this area during the summer season) suggests that absorption might be creating more stable atmospheric situations (by heating aloft layers) and thus preventing clouds formation via convective phenomena and increasing the incoming surface solar radiation. But we did not find any clear evidence of that either (Fig 4c). So the thermodynamic effect of aerosols on clouds inhibition and burn-off might justify the reduction in CCT (mainly at low levels; Fig 4d) and the accompanying increase in RSDS in the southernmost areas. These signals are intensified when we add the indirect aerosols effects, likely due to the removal of aerosols via scavenging processes, which cleans the atmosphere favoring that the solar radiation reaches the surface."

**[293] – The clear sky correlations could be impacted by aerosol-environment co-variability. For instance, dust is associated with dry, hot, cloud-free weather. Those aerosol particles can impact the environment and make it hotter and warmer. Do you think that is at play here?**

Correlations are negative, so the higher the AOD, the lower the RSDS, which is in the opposite direction to the co-variability between dust and cloud-free environments indicated the reviewer. So, if it plays a role, it is not a prevailing role.

**[317] - Evidence to support this argument that cloudiness is the most important?**

The the spatial and temporal correlation values in Supp Fig 6d-i and Supp Fig 7a-f, respectively, are higher between RSDS and CCT than between RSDS and AOD. We have now specifically indicated this argument in the manuscript.

**[Figure 1] – Why is the correlation so low between AOD and RSDS? Even if there is an indirect connection between these two variables via cloudiness, I'm surprised it's so low.**

As described along the manuscript, the variety of indirect mechanisms leading to the different signals in RSDS mask a direct link between AOD and RSDS patterns in Fig 1.

**Anonymous Referee #2**
Authors' response

**Review of the manuscript entitled Sensitivity of surface solar radiation to aerosol-radiation and aerosol-cloud interactions over Europe in WRFv3.6.1 climatic runs with fully interactive aerosols, by S. Jerez et al.**

**This is the revised version of the manuscript, submitted to Geoscientific Model Development, which presents a sensitivity study on the role of dynamic aerosols in regional climate simulations over Europe, carried out with the WRF model.**

**The authors have made substantial efforts to improve the manuscript, and to take into account the different suggestions of the reviewers. The objective of the paper and the results are clarified and better presented. English spelling has also been improved. However, before the publication in GMD, I recommend the following corrections.**

We sincerely thanks the reviewer for agreed to revise our manuscript, especially during this hard times we all are living, and for the constructive comments provided.

**Main comments:**

**- The conclusions about the prevailing of aerosol indirect effects over direct effects should be moderated, since they could be very model-dependent. They could also depend on the choice of the parameterization of cloud-aerosol interactions. More discussions about this aspect should be added in the text.**

Nuanced attributions have been made, and the fact that we rely on a single model configuration has been acknowledged in the discussion section.

**- Many figures are in supplementary material, and are often used in the text of the manuscript. Some sections entirely rely on supplementary figures. At the end, the revised version has only 4 figures in the main text. I think that more figures (not all obviously) should be included in the main text rather than in the supplementary. These figures are essential to better understand the study.**

Following this suggestion, also indicated by the other reviewer, we have now moved the following figures, previously in Supplementary Material, to the main manuscript:

Previously Supp Fig 4 (top panels), now Fig 2: Contribution of each aerosol species to the JJA-mean total surface aerosol concentration in the period 1991-2010. This is now Fig 2 in the revised manuscript.

Previously Supp Fig 5, now Fig 3: Differences between experiments in the RSOT, TAS, RH & CLD JJA climatologies for 1991-2010.  This is now Fig 3 in the revised manuscript.

Previously Supp Fig 6, now Fig 5: Differences between experiments in precipitation-related JJA climatologies for 1991-2010. This is now Fig 4 in the revised manuscript.

**- Section 3.1 : The brief evaluation of AOD (as shown in Figure page 21 in your replies) should be added to the manuscript (at least for JJA). It is an important point to understand the rest of the study. It would come in addition to the brief validation of RSDS in the beginning of section 3.1 (line 229).**

Since the simulated patterns of AOD are already included in Fig 1, we have added a comment on the performance of the simulations at this regards using the paper by Pavlidis et al. (2020) as a reference (see Fig 1 in that paper) and the reviewer's comment on the impact of keeping the 2010 values of anthropogenic emissions along the simulated periods. See the second paragraph of the discussion section.

**- Sections 3.1 and 3.2 could be separated in different subsections in order to avoid too long paragraphs, and to make reading more fluent.**

Done.

**- The authors have mentioned in their replies that they have included in the discussions the justification of keeping constant emissions in future simulations to evaluate the climate change penalty. However I did not find that in the revised version.**

It was mentioned, but probably not clearly enough, true. We have made emphasis on it in the second paragraph of the conclusion section.

**Other comments:**

**- page 1 line 12: please avoid undefined acronyms (IPCC) in the abstract**

Removed.

**- page 1 line 19: Since the period of simulations is 1991-2010, "historical period" would be better than "present-day conditions" (1991 is 30 years ago!). In the whole paper, present could be replaced by historic (or past).**

Ok. We replaced present-day period by historical period throughout the manuscript.

**- page 5 line 121: Is the RRTM radiative scheme used both for shortwave and longwave or longwave only? Please mention the radiative scheme for shortwave in the second option.**

The RRTM scheme is used for both long- and short-wave radiation. Clarified in the text.

**- page 5 line 132: please use subscript characters for chemical species such as NO3 and O3**

Done.

**- page 5 line 142: through instead of trough**

Thanks.

**- page 6 line 148: "Anthropogenic emissions ... were kept unchanged in 148 the simulation periods (we considered the 2010 monthly values)". Anthropogenic emissions have been dramatically reduced between the 1980s and 2010, so keeping 2010 values and comparing simulated aerosols over the whole 1991-2010 period could lead to an underestimation of AOD.**

True (in fact, we observe an underestimation as compared to the MACv2 AOD climatology in the historical period), and to an overestimation during the future period. We have include this appreciation in the discussion section (second paragraph).

**- page 6 line 155: "aerosol optical properties assuming wet particle diameters". Which humidity is used for the calcuation of aerosol optical properties? Is the variation of humidity taken into account on-line in the simulation to allow variations of optical properties (through pre-defined intervals for example)?**

Yes, the variation of humidity is taken into account in the on-line simulations to allow variation of optical properties. We have clarified in the text.

**- page 8 line 216: Could you give the level of significance for the t-test?**

Done.

**- page 13 line 389-390: This point about the autoconversion scheme seems to me very important. Could you explain why you could not use the same scheme, but for example with constant vallue of autoconversion in the case without aerosols?**

It could it be seen as a model limitation, but there is no room for choice. The flag *progn* should be 0 for ARI and 1 for ARCI.